# Spatial distribution of turbulent mixing in the upper ocean of the South China Sea

Xiao-Dong Shang[1], Chang-Rong Liang[1,2], and Gui-Ying Chen[1]

[1]State Key Laboratory of Tropical Oceanography, South China Sea Institute of Oceanology, Chinese Academy of Sciences, Guangzhou 510301, China

[2]University of Chinese Academy of Sciences, Beijing 100049, China

Correspondence to: Xiao-Dong Shang (xdshang@scsio.ac.cn)

**Abstract.** The spatial distribution of the dissipation rate ($\varepsilon$) and diapycnal diffusivity ($\kappa$) in the upper ocean of the South China Sea (SCS) is presented from a measurement program conducted from 26 April to 23 May 2010. In the vertical distribution, the dissipation rates below the surface mixed layer were predominantly high in the thermocline where shear and stratification were strong. In the regional distribution, high dissipation rate and diapycnal diffusivity were observed in the region to the west of Luzon Strait with an average dissipation rate and diapycnal diffusivity of $8.3 \times 10^{-9}$ W kg$^{-1}$ and $2.7 \times 10^{-5}$ m$^2$ s$^{-1}$, respectively, almost one order of magnitude higher than those in the central and southern SCS. In the region to the west of the Luzon Strait, the water column was characterized by strong shear and weak stratification. Elevated dissipation rates ($\varepsilon > 10^{-7}$ W kg$^{-1}$) and diapycnal diffusivity ($\kappa > 10^{-4}$ m$^2$ s$^{-1}$) induced by shear instability occurred in the water column. In the central and southern SCS, the water column was characterized by strong stratification and weak shear and the turbulent mixing was weak. Internal waves and internal tides generated near the Luzon Strait are expected to make dominant contribution to the strong turbulent mixing and shear in the region to the west of the Luzon Strait. The observed dissipation rates were found to scale positively with the shear and stratification, which were consistent with the MacKinnon-Gregg model used for the continental shelf, but different from the Gregg-Henyey scaling used for the open ocean.

## 1 Introduction

Turbulent mixing is a crucial mechanism that controls the distribution of nutrients, sediments, fresh water, and pollutants throughout the water column [Sandstrom and Elliott, 1984]. The magnitude and distribution of diapycnal diffusivity are important for large-scale ocean circulation [Saenko and Merryfield, 2005]. Assuming a balance between vertical advection and vertical diffusion for tracers, Munk [1966] reported that a global average diapycnal diffusivity of $10^{-4}$ m$^2$ s$^{-1}$ is required to maintain gross oceanic stratification and overturning circulation. However, diapycnal diffusivity from turbulent mixing in the open ocean thermocline only ranges from $5 \times 10^{-6}$ to $3 \times 10^{-5}$ m$^2$ s$^{-1}$ [Gregg, 1998; Polzin et al., 1995]. Therefore, it has been argued that elevated turbulent mixing concentrated in rough topographies [Ledwell et al., 2000; Wu et al., 2011] would aid in explaining this discrepancy. In the past decade, elevated diapycnal diffusivities, i.e., $O$ ($10^{-4}$ m$^2$ s$^{-1}$) or higher, have been found in mixing hotspots such as seamounts [Carter et al., 2006; Lueck and Mudge, 1997], ridges [Klymak

et al., 2006a; Lee et al., 2006], and canyons [Carter and Gregg, 2002]. However, these elevated mixing events are highly localized. Whether such topographically enhanced mixing is sufficiently intense or widespread to significantly increase the basin-wide average remains unclear. Using a simple averaging scheme, Kunze and Toole [1997] suggested that topographically induced mixing was insufficient to support a basin-averaged diffusivity of $O$ ($10^{-4}\,\mathrm{m^2\,s^{-1}}$) above a 3000 m depth in the North Pacific.

Compared with the open ocean, less attention has been given to marginal seas. In recent years, observations [Tian et al., 2009] indicated that turbulent mixing in marginal seas could make an important contribution to ocean mixing. The South China Sea (SCS), one of the largest marginal seas of the Pacific, connects to the Pacific through the Luzon Strait. Measurements and numerical simulations [Alford et al., 2015; Chang et al., 2006; Lien et al., 2005] indicated that energetic internal tides and internal waves generated near the Luzon Strait propagate into the SCS and facilitate turbulent mixing. Considerable effort has been put forth to explore the characteristics of turbulent mixing in the SCS. Using fine-scale parameterization, Tian et al. [2009] reported a turbulent mixing distribution along a section from the northern SCS to Pacific. They found that the diapycnal diffusivity in the upper 500 m of the northern SCS reached $O$ ($10^{-5}\,\mathrm{m^2\,s^{-1}}$), almost one order of magnitude larger than that in the Pacific. Yang et al. [2016] explored the turbulent mixing in the SCS with the fine-scale parameterization and found the diapycnal diffusivity in the northern SCS as large as $O$ ($10^{-3}\,\mathrm{m^2\,s^{-1}}$). In addition to these parameterizations, some direct measurements from microstructure profilers are also available. A direct observation of turbulent dissipation was reported by Laurent [2008] who found dissipation rate as high as $10^{-6}\,\mathrm{W\,kg^{-1}}$ over the shelf break of the northern SCS. Lozovatsky et al. [2013] reported a regional mapping of the averaged dissipation rate in the upper pycnocline of the northern SCS and found values in the Luzon Strait as high as $10^{-7}\,\mathrm{W\,kg^{-1}}$. Yang et al. [2014] conducted direct measurements of turbulence along a section across the continental shelf and slope in the northern SCS. Their results show that the averaged dissipation rate over the shelf reached $10^{-7}\,\mathrm{W\,kg^{-1}}$, which is an order of magnitude larger than that over the slope. There is no doubt that these studies have greatly aided our knowledge of turbulent mixing in the SCS. However, the direct microstructure measurements are localized and scattered, with most of them focusing on the northern SCS. Few microstructure measurements have been conducted in the central and southern SCS. Where is the strong turbulent mixing taking place in the SCS and what drives the turbulent mixing are not fully understood. In this work, we present a direct microstructure measurement that covers the upper ocean of the SCS and explore the features and regimes of the turbulent mixing. Energy sources for the turbulent mixing are also discussed. In addition, there is a lack of studies assessing parameterizations in the SCS. In order to estimate the turbulent mixing without microstructure measurements, we assess two fine-scale parameterizations with microstructure data and investigate which one works better and why it works better. Fine-scale parameterizations are aimed at reproducing the dissipation rate in terms of more easily observed or modeled quantities, such as stratification and shear. Generally, microstructure measurements are fewer and more difficult than the fine-structure measurements (e.g., CTD and ADCP measurements), especially microstructure measurements in deep sea. Therefore, to study the spatial and temporal distribution of turbulent mixing, researchers often resort to the fine-scale parameterizations [Jing and Wu, 2010; Tian et al., 2009; Wu et al., 2011]. In addition, fine-scale parameterizations would provide reference for

modelers. Shelf sea models have success in reproducing the water column structure in seasonally stratified shelf seas[Holt and Umlauf, 2008; Simpson and Bowers, 1981]. However, models need to calibrate a background mixing level to correctly predict the water column structure [Rippeth, 2005]. The requirement of calibration reduces the success of models on shelf-wide scales since differing forcing mechanisms and mixing processes require specific methods and levels of tuning. This presents a clear challenge to oceanographic models. Before the water column structure in shelf seas can be modeled realistically, the distribution of mixing must be established and the major mixing processes parameterized. Confidence in future predictions is therefore dependent on ocean turbulence model that can be validated against observed mixing or parameterized mixing, but not on the calibration of a background mixing level. We begin in section 2 with a description of our measurements and methods. In section 3 we explore the features and regimes of the turbulent mixing, and assess two fine-scale parameterizations. We discuss the turbulent mixing and fine-scale parameterizations in section 4. A summary of our results is presented in section 5.

## 2 Measurements and Methods

The field experiment was performed from 26 April to 23 May 2010 (local time) prior to the South China Sea summer monsoon (SCSSM) onset. A total of 82 stations were conducted in the experiment (Fig. 1a). Direct turbulence measurements were collected with the Turbulence Ocean Microstructure Acquisition Profiler (TurboMAP), TurboMAP is a quasi-free-falling instrument that measures turbulent parameters ($\partial u/\partial z$ and $\partial T/\partial z$), bio-optical parameters (in vivo fluorescence and backscatter), and hydrographic parameters (conductivity, temperature, and depth) [Wolk et al., 2002]. TurboMAP carries seven environmental sensors and a three-axis accelerometer that measures tilt and vibrations. The turbulent velocity fluctuations are measured with two standard shear probes. Conductivity ($C$) and temperature ($T$) are measured with a combined $C–T$ sensor consisting of a platinum wire thermometer and an inductive conductivity cell. Depth is measured with a semiconductor strain gauge pressure transducer, and the instrument's sinking velocity is computed from the rate of change of the pressure signal. All sensors are sampled at a rate of 256 Hz. TurboMAP was deployed at a speed of 0.5-0.7 m s$^{-1}$ and the maximum deployment depth was approximately 800 m. It took 10-30 min to complete each profile at shallow stations and approximately one hour at deep stations. Continuous time series of velocity at 5-min intervals and 16-m vertical spacing between 38 and 982 m were obtained from a shipboard acoustic Doppler current profiler (ADCP). At stations where the water depth was more than 982 m, the current velocity cannot be referenced to the sea floor. The movement of the ship was determined from GPS data and absolute value of current velocity was estimated. CTD casts were conducted to provide measurements of temperature and salinity for comparison. At stations where the water depth was less than 800 m, CTD was deployed to 5 m above the seafloor. At stations where the water depth was larger than 800 m, the maximum deployment depth of CTD ranged from 800 to 1500 m. Data obtained from six moorings (Fig. 1, yellow squares) were used to perform a brief analysis of the internal wave field in the SCS. Moorings 1-3 were deployed over the continental shelf/slope and moorings 4-6 were deployed in the deep basin. More information regarding the moorings is given in Table 1.

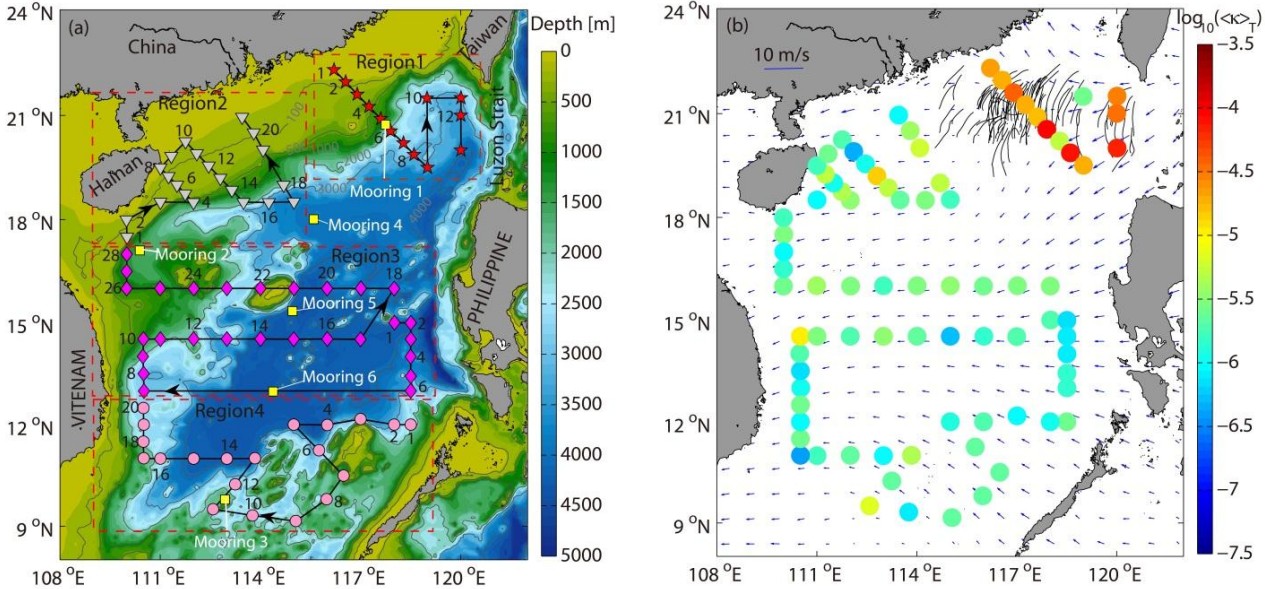

Fig. 1. (a) Bottom topography of the SCS and observation stations (symbols). Red stars indicate the stations located in region 1. Gray triangles indicate the stations located in region 2. Magenta diamonds indicate the stations located in region 3. Pink dots indicate the stations located in region 4. Station numbers (i.e., 1, 2, 4…) are indicated in each region. The arrows indicate the order of the measurement. The yellow squares indicate the locations of the moorings. (b) Spatial distribution of $<\kappa>_T$ (color dots). The blue vector gives the averaged 10 m wind speed during the cruises. Black curves in the northern SCS are internal wave packets derived from satellite images by Zhao et al. [2004].

Figure 2a shows depth profile of shear $\partial u/\partial z$. At depths of 190-200 m, the shear signal shows variations with peak levels around 0.6 s$^{-1}$, corresponding to dissipation rates of $10^{-7}$ W/kg. The velocity shear decreases below 200 m to peak values of 0.02 s$^{-1}$, corresponding to dissipation rates of $10^{-10}$ W/kg. Dissipation spectra $\psi(k)$ computed from the shear signal in Fig. 2a are shown in Figs. 2c-2g along with the corresponding scaled Nasmyth universal spectra [Nasmyth, 1970]. The shape of the measured spectra agrees well with the universal spectrum except in the wavenumber regions affected by vibration noise caused by the strumming of the suspension wires in the flow [Wolk et al., 2002]. The spectra are computed using Welch's averaged periodogram method with an FFT length of 2 m, corresponding to a consecutive segment of approximately 1700 data points. The dissipation rates $\varepsilon$ based on the measured spectra range from $10^{-10}$ W kg$^{-1}$ to $10^{-7}$ W kg$^{-1}$, and they are computed by integrating the measured shear spectrum

$$\varepsilon = 7.5\nu \left\langle \left(\frac{\partial u}{\partial z}\right)^2 \right\rangle = 7.5\nu \int_{k_1}^{k_2} \psi(k)\, dk,$$

where $\nu$ is the kinematic viscosity and $< >$ denotes the spatial average. $k_1$ and $k_2$ are the integration limits. The lower integration limit $k_1$ is set to 1 cpm and the upper limit $k_2$ is the highest wavenumber that is not contaminated by vibration noise. Energy density in the low wavenumber area around 1 cpm is not well estimated because of the limited length of the

data segments and the length of the profiler itself. The noise level of the TurboMAP profiler is $\varepsilon \sim 10^{-10}$ W/kg [Matsuno and Wolk, 2005; Wolk et al., 2002]. Diapycnal diffusivity [Osborn, 1980] was calculated based on the dissipation rate $\varepsilon$ and stratification $N$ using

$$\kappa = \Gamma \varepsilon / N^2,$$

where the mixing efficiency $\Gamma$ is set to 0.2 [Oakey, 1982]. The shear variance, $S^2 = (\Delta \bar{U}/\Delta z)^2 + (\Delta \bar{V}/\Delta z)^2$, was calculated with $\Delta z = 16$ m, where $\bar{U}$ and $\bar{V}$ are the zonal and meridional components of the mean horizontal velocity obtained from the shipboard ADCP, respectively. The mean velocity is averaged over the time intervals of the TurboMAP measurements.

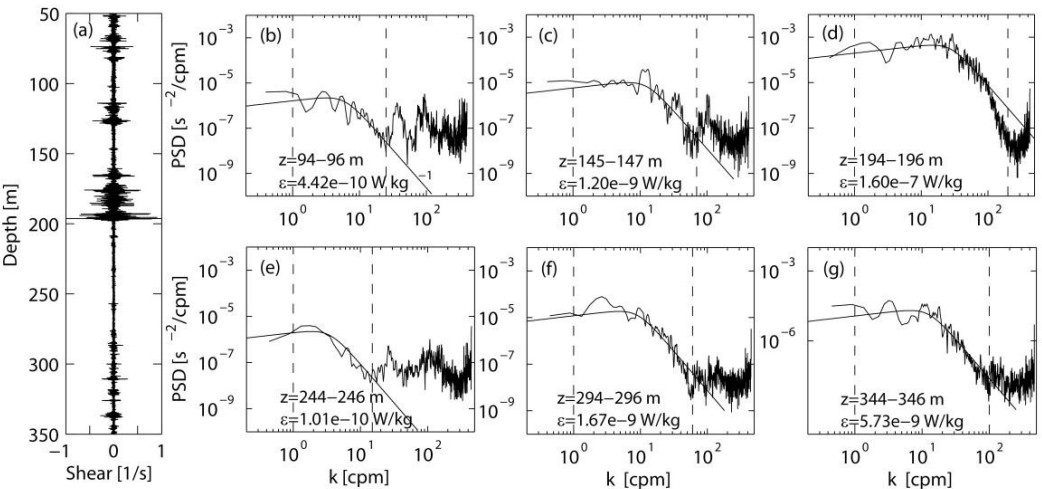

Fig. 2. Examples of (a) micro-shear and (b-g) shear spectra at different depths. The integration bounds (vertical dashed lines) and Nasmyth spectra (smooth curves) are shown.

**Table 1.** Information about the moorings.

| Mooring | Latitude (°N) | Longitude (°E) | Water Depth (m) | Measurement Depth Range (m) | Measurement Duration (d/m/yr) | Time Interval (min) | Bin Size (m) |
|---|---|---|---|---|---|---|---|
| Mooring 1 | 20.74 | 117.75 | 1260 | 13~454 | 01/08/14~27/09/14 | 2 | 16 |
| Mooring 2 | 17.10 | 110.39 | 1410 | 6~478 | 04/05/09~04/09/10 | 60 | 8 |
| Mooring 3 | 9.79 | 112.74 | 1680 | 40~416 | 25/05/09~10/11/10 | 60 | 8 |
| Mooring 4 | 18.01 | 115.60 | 3790 | 60~370 | 09/04/98~05/10//98 | 60 | 10 |
| Mooring 5 | 15.34 | 114.96 | 4265 | 30~270 | 07/10/98~11/04//99 | 60 | 10 |
| Mooring 6 | 12.98 | 114.38 | 4370 | 30~270 | 09/10/98~12/04//99 | 60 | 10 |

## 3 Results

### 3.1 Water mass properties

Intrusion of water from the Pacific can influence the evolving water properties in the SCS. It has been confirmed by in situ measurements and models [Shaw, 1991; Wu and Hsin, 2012] that there is a strong intrusion of water from the Pacific into the SCS through the Luzon Strait. Two well-defined water masses are active in this process [Qu et al., 2000]: high-salinity North Pacific Tropical Water (NPTW) and low-salinity North Pacific Intermediate Water (NPIW). For simplicity, we divide the observations into four regions (Fig. 1): region 1 is located to the west of the Luzon Strait, region 2 is located to

the northeast of Hainan Island, region 3 is located in the central SCS, and region 4 is located in the southern SCS. Fig. 3 shows the $T$-$S$ curves of the SCS and western Pacific. Temperature and salinity data in the western Pacific ($18.5^{o}$ N-$22.5^{o}$ N, $124.5^{o}$ E-$128.5^{o}$ E) were obtained from the World Ocean Database 2013 (http://www.nodc.noaa.gov/OC5/woa13/woa13 data.html). The $T$-$S$ curve in the western Pacific shows a reversed 'S' shape with NPTW and NPIW clearly identified (Fig. 3, black dashed curve). NPTW and NPIW correspond to the maximum salinity layer at $\sigma_\theta \in (22.5$-$25.5)$ kg m$^{-3}$ and minimum

salinity layer at $\sigma_\theta \in (25.5$-$27.5)$ kg m$^{-3}$, respectively. In the maximum salinity layer (22.5-25.5 kg m$^{-3}$), the water column in region 1 had a salinity maximum of 34.8 psu that approaches the maximum value of the NPTW. Salinity decreased gradually from the Luzon Strait to the Hainan Island (region 2) and to the central and southern SCS (region 3 and region 4). This trend is reversed in the minimum salinity layer (25.5-27.5 kg m$^{-3}$), where the salinity slightly increased from the Luzon Strait to Hainan Island and to the central and southern SCS. The salinity minimum in the Pacific was found to be lower than that in

the SCS. Reverse S shape becomes remarkably weak from the northern SCS to the southern SCS, a change to which turbulent mixing occurring in the SCS might have made a significant contribution.

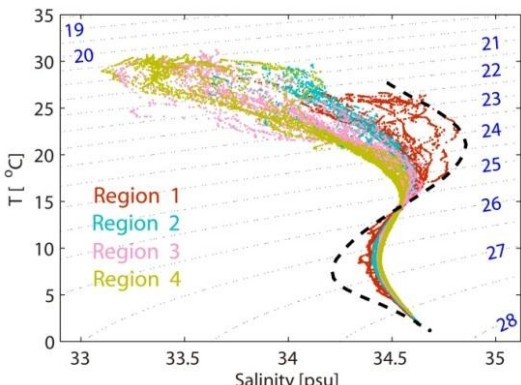

Fig. 3. Relation of potential temperature vs. salinity (with the potential density $\sigma_\theta$ in kg/m$^3$ contours overlaid) of all stations. The black dashed curve shows the relation for potential temperature vs. salinity of the western Pacific for reference.

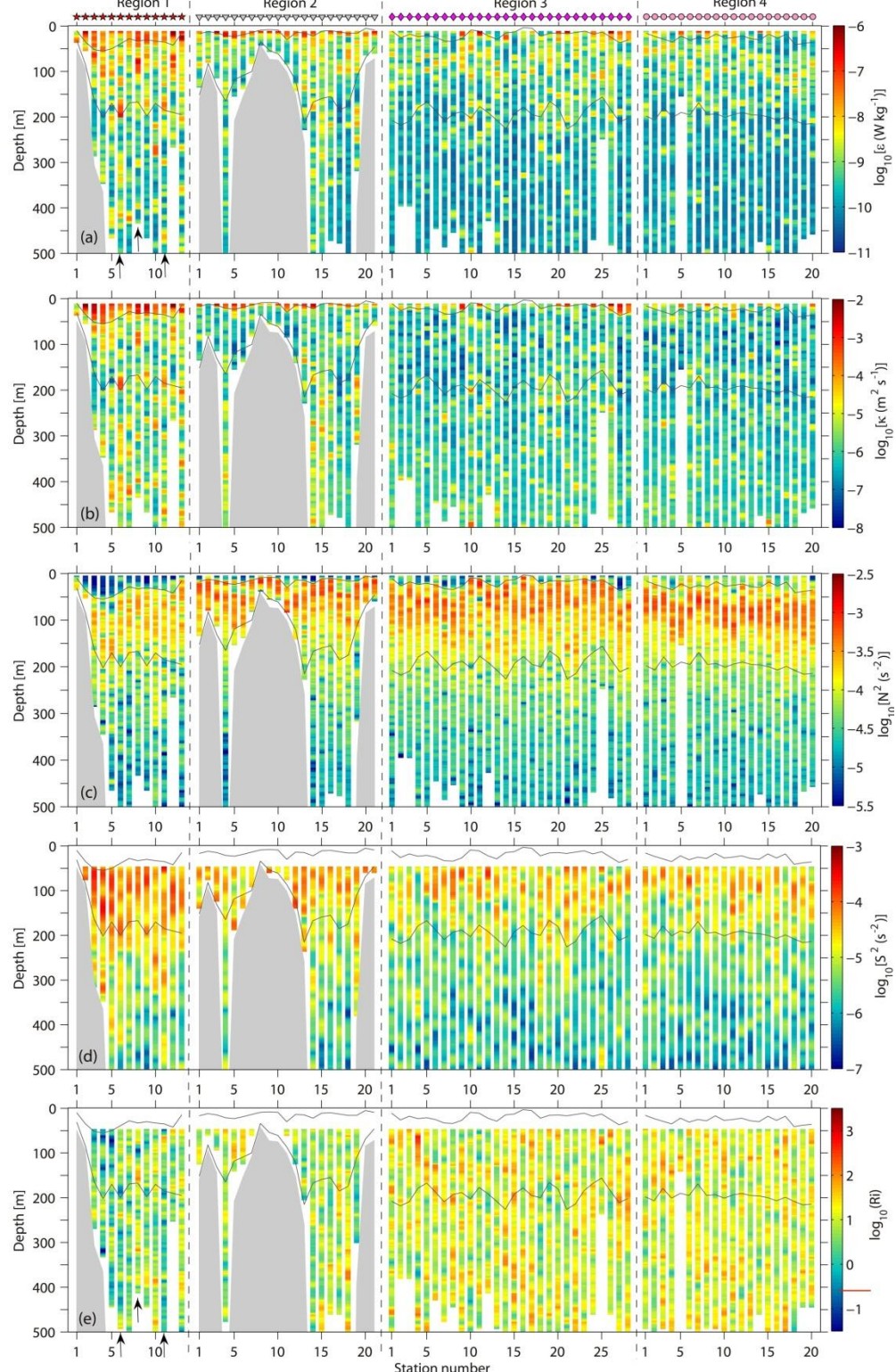

Fig. 4. (a) Dissipation rate ($\varepsilon$), (b) diapycnal diffusivity ($\kappa$), (c) buoyancy frequency squared ($N^2$), (d) shear variance ($S^2$), and (e) Richardson number ($Ri$) from all of the stations. The gray shading indicates the bathymetry. In (a)-(e) the boundaries of the thermocline are indicated (gray curves). The red line on the color bar of (e) represents $Ri$ =0.25. The vertical dashed lines divide the stations into four regions with the symbols (red stars, gray triangles, magenta diamonds, and pink dots) shown at the top of (a). These symbols correspond to the station symbols in Fig. 1a.

## 3.2 Microstructure Measurements

Figure 4a shows the distribution of the dissipation rate with the thermocline boundaries overlain. Different criteria have been used to define the top of the thermocline ($z_t$) in terms of either temperature or density. Here, we defined the top of the thermocline as the depth at which the potential temperature change from the surface temperature is 0.5 ˚C. The bottom of the thermocline ($z_b$) is defined as the depth at which the temperature gradient is equal to 0.05 ˚C m$^{-1}$. The surface mixed layers are slightly deep in region 1 compared with the other regions. The average depths of the surface mixed layer in regions 1-4 are 35.2, 14.7, 19.4, and 26.8 m, respectively.

In the surface mixed layer, strong turbulence was accompanied by high dissipation rates (Fig. 4a), which may be attributed to various factors, such as wind stirring, buoyancy flux, and surface waves. Below the surface mixed layer, high dissipation rates (Fig. 4a) were observed in the thermocline, with the average $\varepsilon$ in the thermocline reaching $4.6 \times 10^{-9}$ W kg$^{-1}$, which was five times larger than the value of $8.2 \times 10^{-10}$ W kg$^{-1}$ below the thermocline. Strong shear (Fig. 4d) also occurred in the thermocline with an averaged $S^2$ in the thermocline of $3.3 \times 10^{-5}$ s$^{-2}$, which was five times larger than that below the thermocline ($6.5 \times 10^{-6}$ s$^{-2}$). The strong spatial correlation between dissipation and shear implies that shear played an important role in driving the dissipation. Contrary to the dissipation rates, the diapycnal diffusivities (Fig. 4b) in the thermocline were slightly weaker than that below the thermocline. The high diapycnal diffusivities below the thermocline were mainly due to the relatively weak stratification (Fig. 4c). The average $N^2$ below the thermocline was $8.4 \times 10^{-5}$ s$^{-2}$, four times smaller than the value of $3.4 \times 10^{-4}$ s$^{-2}$ in the thermocline.

Turbulent mixing in region 1 displayed a different feature from that of the other regions. In region 1, turbulence was more active than that in other regions, with the maximum dissipation rate reaching $10^{-6}$ W kg$^{-1}$ (Fig. 4a) and the maximum diapycnal diffusivity exceeding $10^{-3}$ m$^2$ s$^{-1}$ (Fig. 4b). In addition, region 1 had weak stratification but strong shear compared with other regions (Figs. 4c and 4d). Most of the water column in region 1 was occupied by Richardson number of order 1, almost two orders of magnitude smaller than that in the other regions (Fig. 4e). Richardson number $Ri = N^2/S^2$ was estimated following MacKinnon and Gregg [2005]. 2-m buoyancy frequency and 16-m shear were used in the calculation. The resolutions of shear used in previous literatures range from 2 m to 16 m [MacKinnon and Gregg, 2003b; 2005; van der Lee and Umlauf, 2011; Xie et al., 2013; Yang et al., 2014]. High resolution of shear (2-4 m) was used on the shelf area to resolve small scale internal waves and low resolution of shear (8-16 m) was often used in deep water to cover more water depth. Although Richardson number calculated on 16-m shear might be underestimated, it does not affect the comparison of Richardson number in different regions too much. Another distinguishing feature in region 1 is that some turbulent patches

with elevated dissipation rates ($\varepsilon > 10^{-7}$ W kg$^{-1}$) and diapycnal diffusivities ($\kappa > 10^{-4}$ m$^2$ s$^{-1}$) were observed in the water column. These turbulent patches often occurred at depths where the Richardson number was below 0.25; for example, station 6 (between 175 and 195 m), station 8 (between 80 and 100 m), and station 11 (between 175 and 195 m) (indicated by the arrows in Fig. 4), which suggests that elevated dissipation rates and diapycnal diffusivities in the turbulent patches are likely to result from shear instability. More detail regarding the shear instability will be discussed in the following text. Compared with region 1, turbulent mixing in regions 2-4 was relatively weak, with an average $\varepsilon$ and $\kappa$ in the upper 500 m (not including the surface mixed layer) of $1.1 \times 10^{-9}$ W kg$^{-1}$ and $3.7 \times 10^{-6}$ m$^2$ s$^{-1}$, respectively. These two values are almost one order of magnitude smaller than those ($\varepsilon \sim 8.3 \times 10^{-9}$ W kg$^{-1}$ and $\kappa \sim 2.7 \times 10^{-5}$ m$^2$ s$^{-1}$) in region 1. Weak turbulent mixing in regions 2-4 is likely to be associated with the strong stratification and weak shear. $N^2$ (Fig. 4c) was greater than $S^2$ (Fig. 4d) in regions 2-4 with most of the water column occupied by large Richardson number ($Ri > 10$) (Fig. 4e).

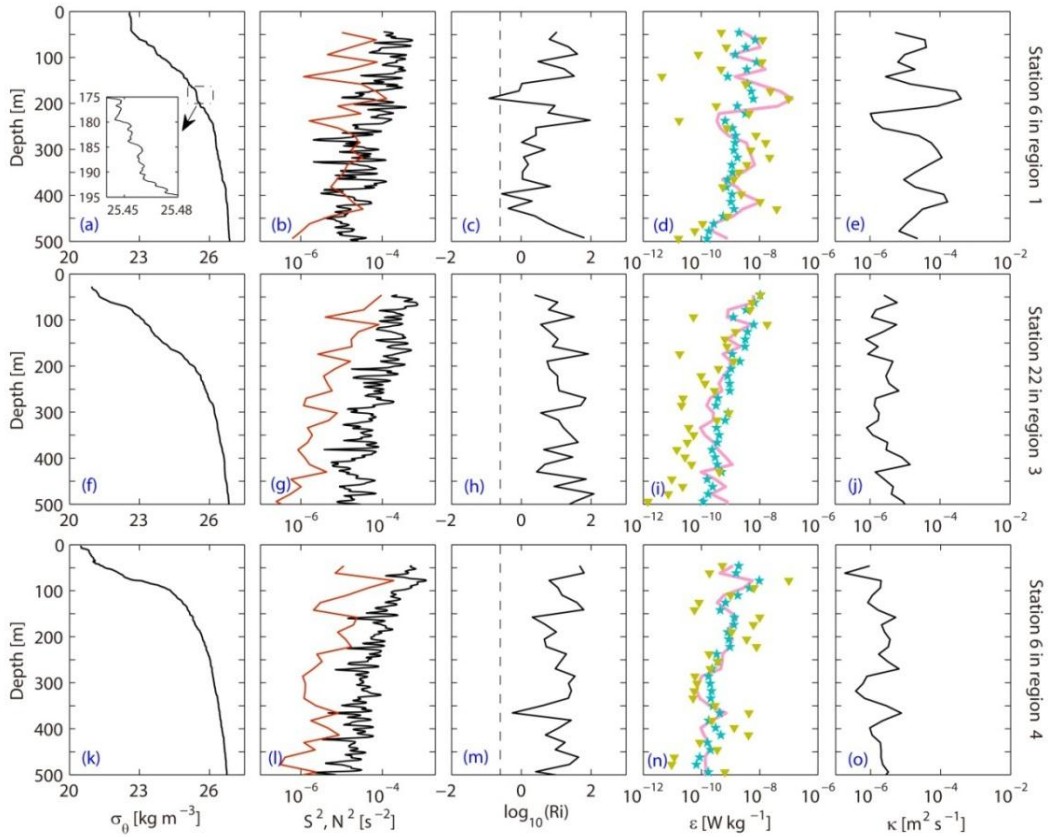

Fig. 5. From top to bottom, three sets of profiles are from station 6 in region 1 (top panels), station 22 in region 3 (middle panels), and station 6 in region 4 (bottom panels). For each station, quantities plotted are (from left to right) potential density, shear variance (red) and buoyancy frequency squared (black), Richardson number (the vertical line indicates $Ri = 0.25$), observed (pink curves) MG model (stars) and GH model (triangles) dissipation rates, and observed diapycnal diffusivity. The

observed dissipation rate and diapycnal diffusivity have been vertically averaged over the 16 m ADCP bins. The inset in (a) enlarges the density profile to show the overturns.

To further understand the changing pattern of turbulence in the SCS, we now look in detail at the profiles of various quantities at three stations in different regions (Fig. 5): station 6 was from region 1, station 22 was from region 3, and station 6 was from region 4. At station 6 in region 1 (Figs. 5a-5e), the shear variance was slightly smaller than the buoyancy frequency squared over most of the water column (Fig. 5b). However, the shear variance exceeded the buoyancy frequency squared at some depths, for example, the shear variance was greater than the buoyancy frequency squared at depth of 185 m, pushing the Richardson number below 0.25, which implies shear instability (Fig. 5c). Small overturns were also found in the density profile at depths of 175 to 195 m (Fig. 5a, the inset). The dissipation rates (Fig. 5d) and diapycnal diffusivities (Fig. 5e) in the corresponding depths (175-195 m) were elevated by more than one order of magnitude with the diapycnal diffusivities reaching $5.0 \times 10^{-4}$ m$^2$ s$^{-1}$, one to two orders of magnitude higher than the levels in open ocean thermocline. The dissipation rates induced by shear instability contributed significantly to the turbulent mixing in the water column. Nearly 45% of the total dissipation rates in the upper 500 m (not including the surface mixed layer) was contributed by the elevated dissipation rates from the turbulent patch. The second and third sets of profiles were from region 3 (Figs. 5f-5j) and region 4 (Figs. 5k-5o), respectively. The buoyancy frequency squared was higher than the shear variance (Figs. 5g and 5l), and no Richardson numbers below 0.25 were observed (Figs. 5h and 5m). The water column was occupied by dissipation rates ranging from $10^{-10}$ to $10^{-9}$ W kg$^{-1}$ (Figs. 5i and 5n) and diapycnal diffusivities of $10^{-6}$ to $10^{-5}$ m$^2$ s$^{-1}$ (Figs. 5j and 5o), comparable to the levels in open ocean thermocline.

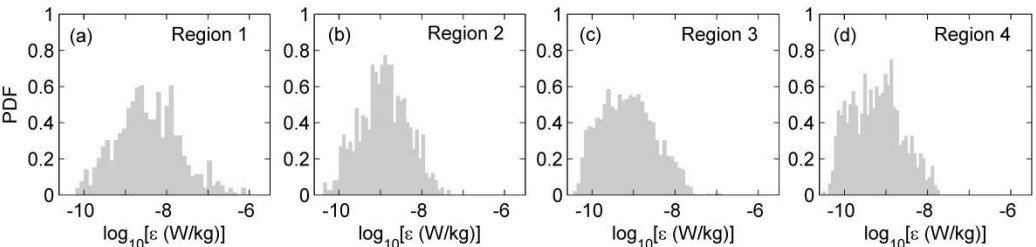

Fig. 6. Probability density functions of dissipation rates in (a) region 1, (b) region 2, (c) region 3, and (d) region 4.

The above analysis indicates that high dissipation rates mainly occurred in the thermocline and the distribution of thermocline dissipation was spatially non-uniform. Turbulent mixing in the thermocline can be driven by various factors, such as surface wind, internal waves, and internal tides. In order to find out whether the turbulent mixing in the thermocline is driven by a single forcing or multiple forcing, we explore the probability density function (PDF) of dissipation rates estimated from non-parametric PDF estimator. The PDFs of dissipation rates (Fig. 6) in the four regions do not show sharp shapes with a single significant peak. Instead, they show flat shapes with multiple peaks, especially the PDFs in regions 1 and 4, which suggests that the turbulent mixing in the thermocline is driven by multiple forcing. To further explore the

energy sources to the thermocline dissipation, we calculate the averaged dissipation rate $<\varepsilon>_T$ and the averaged diapycnal diffusivity $<\kappa>_T$ in the thermocline. $<\varepsilon>_T$ and $<\kappa>_T$ are given by

$$\langle\varepsilon\rangle_T = 1/(z_b-z_t) \int_{z_b}^{z_t} \varepsilon dz \text{ and } \langle\kappa\rangle_T = 1/(z_b-z_t) \int_{z_b}^{z_t} \kappa dz,$$

where $z_b$ and $z_t$ are the bottom and top of the thermocline, respectively. The dissipation rates and diapycnal diffusivities affected by the surface mixed layer were excluded before calculating $<\varepsilon>_T$ and $<\kappa>_T$. Fig. 7b shows the averaged dissipation rate in the thermocline. $<\varepsilon>_T$ displayed a decreased trend toward the south from $O(10^{-8} \text{ W kg}^{-1})$ in region 1 to $O(10^{-9} \text{ W kg}^{-1})$ in region 4. In region 1, $<\varepsilon>_T$ ranged from $1.8\times10^{-9}$ to $5.0\times10^{-8} \text{ W kg}^{-1}$ with a mean value of $1.8\times10^{-8} \text{ W kg}^{-1}$, which was 7 times, 9 times, and 12 times higher than the mean values of region 2 ($2.5\times10^{-9} \text{ W kg}^{-1}$), region 3 ($2.1\times10^{-9} \text{ W kg}^{-1}$), and region 4 ($1.5\times10^{-9} \text{ W kg}^{-1}$), respectively. Elevated $<\kappa>_T$ was also observed in region 1 (Fig. 7c). The average $<\kappa>_T$ in region 1 was $3.5\times10^{-5} \text{ m}^2 \text{ s}^{-1}$, which was an order of magnitude greater than the values of region 2 ($3.3\times10^{-6} \text{ m}^2 \text{ s}^{-1}$), region 3 ($2.2\times10^{-6} \text{ m}^2 \text{ s}^{-1}$), and region 4 ($2.1\times10^{-6} \text{ m}^2 \text{ s}^{-1}$). One prominent feature in the northern SCS is that the mean of $<\kappa>_T$ in region 1 was 11 times higher than the value in region 2 while the mean of $<\varepsilon>_T$ in region 1 was only 7 times higher than that the value in region 2. This difference mainly resulted from the weak stratification in region 1 (Fig. 4c).

Microstructure measurements at different stations were taken at different time and the measure time might be one of the factors that affect the variability of $<\varepsilon>_T$ and $<\kappa>_T$. Strong turbulent mixing generally occurs during spring tides [Peters and Bokhorst, 2000]. Thus it is possible that microstructure measurements in region 1 were taken during spring tides and those in regions 2-4 were taken during neap tides, and the elevated turbulent mixing in region 1 may result from different measure time. To rule out this possibility, we obtained the barotropic tides from the global inverse tide model (TPXO) [Egbert and Erofeeva, 2002], which give us the time information of spring-neap tides during the period of observation. Only the barotropic tides at $18^{\circ}$N, $114^{\circ}$E were extracted because the bias in the arrival of spring-neap tides in different locations of the SCS is small (no longer than 3 hours, not shown). The 14-day spring-neap cycles were well-represented in the extracted barotropic tides (Fig. 7d). A comparison of $<\varepsilon>_T$ and $<\kappa>_T$ to the extracted tides suggests that elevated turbulent mixing in region 1 were not attributed to the measure time, for example, stations in regions 1 and 3 spanned neap and spring tides (see Fig. 7d, stars and diamonds), but the averaged $<\varepsilon>_T$ and $<\kappa>_T$ in region 1 were still an order of magnitude greater than the values in region 3 (Figs. 7b and 7c).

Surface wind is an important energy source for the turbulence in the ocean [Brainerd and Gregg, 1993; Burchard and Rippeth, 2009; Matsuno et al., 2005; Shay and Gregg, 1986], which indirectly enhances the turbulence in thermocline through inertial-gravity wave motion generated by surface wind stress. To find out whether surface wind affects the turbulence in the thermocline significantly, we estimate the wind energy flux. The wind energy flux [Yang et al., 2014] at a height of 10 m, $E_{10}$, is given by $E_{10} = \rho_a C_D U_{10}^3$, where $\rho_a$=1.2 kg m$^{-3}$ is the air density, $C_D$ is the drag coefficient with a value of $1.14\times10^{-3}$ [Large and Pond, 1981], and $U_{10}$ is the wind speed at 10 m. The wind speed data during the observation come from the European Centre for Medium-Range Weather Forecasts (http://apps.ecmwf.int/datasets/data/interim-full-daily/levtype=sfc/). The variability of $E_{10}$ is shown in Fig. 7a. Winds were light ($U_{10} < 9$ m s$^{-1}$) at all the stations with $E_{10}<$

1.0 W m$^{-2}$ except for stations 11-13 in region 1. The influence of wind stress on the variability of turbulence in thermocline

was small, as one can see from Figs. 7a-7c that the variability of $<\varepsilon>_T$ and $<\kappa>_T$ did not follow the variability of $E_{10}$. The values of $E_{10}$ ($10^{-1}$ W m$^{-2}$) at stations 1-9 in region 4 were an order of magnitude larger than that at stations 1-7 in region 1 while the values of $<\varepsilon>_T$ and $<\kappa>_T$ at stations 1-9 in region 4 were almost an order of magnitude smaller than that at stations 1-7 in region 1. Evidence also can be found from the comparison between $<\kappa>_T$ and averaged wind speed during the cruises (Fig. 1b). The average winds were evenly distributed over the SCS, which is significantly different from the spatial

distribution of $<\kappa>_T$. These observations suggest that the contribution of surface winds to the observed strong turbulence in region 1 was small. Measurements from Matsuno and Wolk [2005] also indicates that contribution of surface winds to the turbulence below the surface mixing layer was small during light winds and only when the wind speed reached 10 m s$^{-1}$ would wind stirring made a notable contribution to the turbulence below the surface mixing layer.

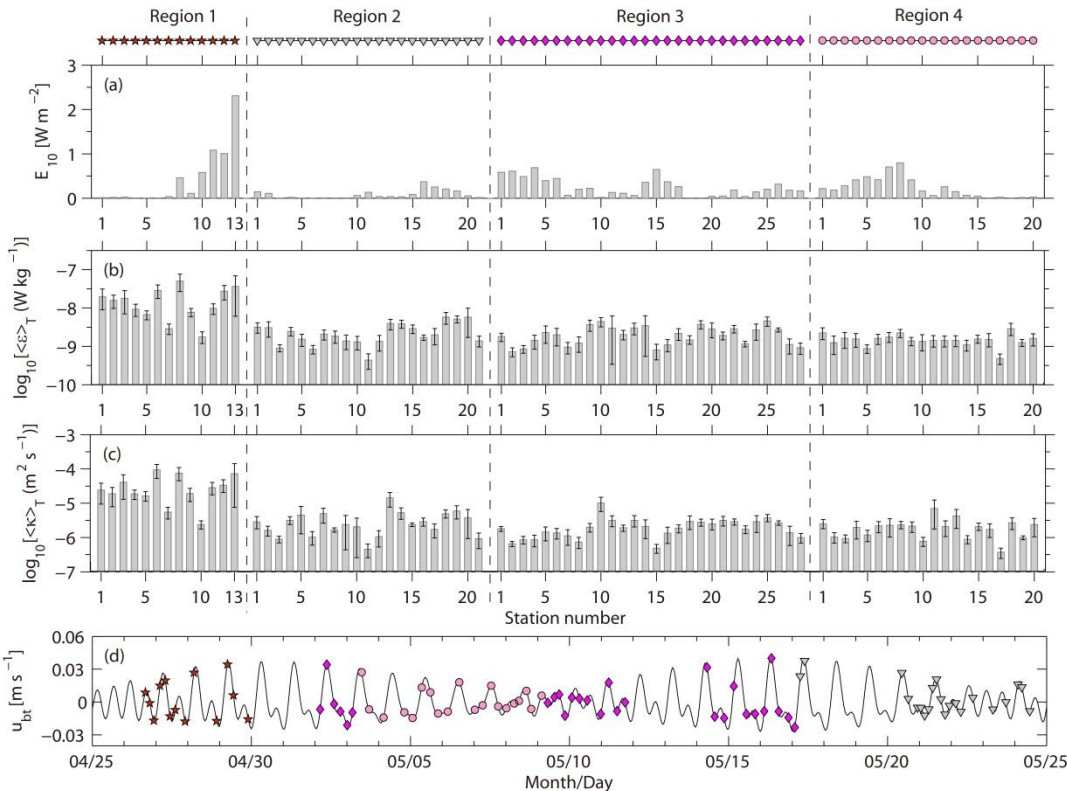

Fig. 7. (a) Wind energy flux $E_{10}$ for each station during the TurboMAP measurement. (b) The average dissipation rate $<\varepsilon>_T$ and (c) average diapycnal diffusivity $<\kappa>_T$ in the thermocline. The vertical bars in (b) and (c) indicate the 95% bootstrapped confidence interval. The vertical dashed lines divide the stations into four regions with symbols (red stars, gray triangles, magenta diamonds, and pink dots) shown at the top of (a). These symbols correspond to the station symbols in Fig. 1a. (d) Time series of the barotropic tidal velocity ($u_{bt}$) predicted from TPXO 7.1 with the station symbols overlain.

Internal waves and internal tides are the candidates that contribute to the elevated turbulent mixing in region 1. It is known that internal waves can provide large amounts of energy for turbulence in the ocean [Alford et al., 2015]. Internal waves are unevenly distributed throughout the SCS. Most of internal waves and internal tides originate in Luzon Strait and propagate northwestwards through the deep water zone near Luzon Strait to the continental shelf [Guo and Chen, 2014; Klymak et al., 2006b; Lien et al., 2005; Ramp et al., 2004; Zhao, 2014; Zhao et al., 2004]. Internal wave packets derived

from satellite images by Zhao et al. [2004] are shown in Fig. 1b for reference. Most of the internal wave packets occurred on the continental shelf in region 1 where $<\kappa>_T$ can be $10^{-5}$-$10^{-4}$ $m^2$ $s^{-1}$, almost an order of magnitude greater than that on the adjacent continental shelf in region 2. Report based on mooring data [Lien et al., 2014] indicates that internal waves would induce strong shear during propagation. Strong shear were also found in region 1 in our measurement (Fig. 4d). These observations suggested that internal waves and internal tides generated near Luzon Strait are expected to make a dominant

contribution to the elevated turbulence in region 1.

     Here we provide a brief summary for this section. The observation indicates that strong turbulent mixing was concentrated in region 1, to which stratification and shear variance made significant contribution. The water column in region 1 had weaker stratification but stronger shear than those in other regions. In this condition, shear instability events occasionally occurred in region 1 and produced elevated dissipation and diapycnal diffusivity. In regions 2-4, instead, the

water column was characterized by weak shear and strong stratification. Shear was no longer sufficiently high to produce subcritical Richardson numbers. Thus the resulting turbulence was weak. Analysis indicates that the spatial distribution of turbulent mixing with large $<\varepsilon>_T$ and $<\kappa>_T$ concentrated in region 1 does not result from the measure time or surface winds. The energetic internal waves and internal tides generated near the Luzon Strait are expected to make a dominant contribution to create this mixing pattern.

**3.3 Parameterizations of turbulence**

     In this section we evaluate two models for parameterizing the dissipation rate in terms of more easily observed or modeled quantities, such as stratification and shear. One wave-wave interaction parameterization [Gregg, 1989; MacKinnon and Gregg, 2003a] in the open ocean is the Gregg-Henyey scaling (known as the GH model) given by

$$\varepsilon_{GH} = \alpha_0 \left[ f cosh^{-1} \left( \frac{N_0}{f} \right) \right] \left( \frac{S^4}{S_{GM}^4} \right) \left( \frac{N^2}{N_0^2} \right) \text{ and}$$

$$S_{GM}^4 = \beta_0 \left( \frac{N^2}{N_0^2} \right)^2,$$

where $\alpha_0$=1.8×$10^{-6}$ J $kg^{-1}$, $f$ is the Coriolis frequency, $S$ is the low-frequency/low-mode resolved shear, $N_0$ is a reference

buoyancy frequency, $cosh^{-1}$ denotes inverse hyperbolic cosine function, and $\beta_0$=1.66×$10^{-10}$ $s^{-4}$. Another analytical model [MacKinnon and Gregg, 2003a] is the MacKinnon-Gregg model (known as the MG model) given by

$$\varepsilon_{MG} = \varepsilon_0 \left( \frac{N}{N_0} \right) \left( \frac{S}{S_0} \right),$$

where $S_0=N_0=5.1\times10^{-3}$ s$^{-1}$ and $\varepsilon_0$ is an adjustable constant that gives the model dissipation rate the same cruise average as the observational data. The adjustable constant $\varepsilon_0$ shows great variability in different regions and seasons, spanning from $10^{-10}$ W kg$^{-1}$ to more than $10^{-8}$ W kg$^{-1}$ [MacKinnon and Gregg, 2005; Palmer et al., 2008; van der Lee and Umlauf, 2011; Xie et al., 2013]. This regional and temporal variability of $\varepsilon_0$ strongly suggests the importance of different physical processes on setup and maintenance of the background levels of turbulent dissipation. Here, we assess the two models for parameterization of the turbulence in the northern SCS (dissipation data are from the stations in region 1 and region 2), central SCS (dissipation data are from the stations in region 3), and southern SCS (dissipation data are from the stations in region 4). Different values of parameter $\varepsilon_0$ are selected for the parameterizations due to their different mixing backgrounds: $\varepsilon_0=1.65\times10^{-9}$ W kg$^{-1}$ for the northern SCS, $\varepsilon_0=0.96\times10^{-9}$ W kg$^{-1}$ for the central SCS, and $\varepsilon_0=0.50\times10^{-9}$ W kg$^{-1}$ for the southern SCS. All of the data affected by the surface mixed layers or bottom mixed layers was excluded for the parameterizations. To reduce the bias introduced by the different vertical resolutions of the shear and stratification data, 16-m buoyancy frequency and 16-m shear were used in the parameterization, namely, density was first interpolated onto the ADCP grid and $N^2$ was computed from finite differencing. Accordingly, the dissipation rates were vertically averaged over the 16-m ADCP bins.

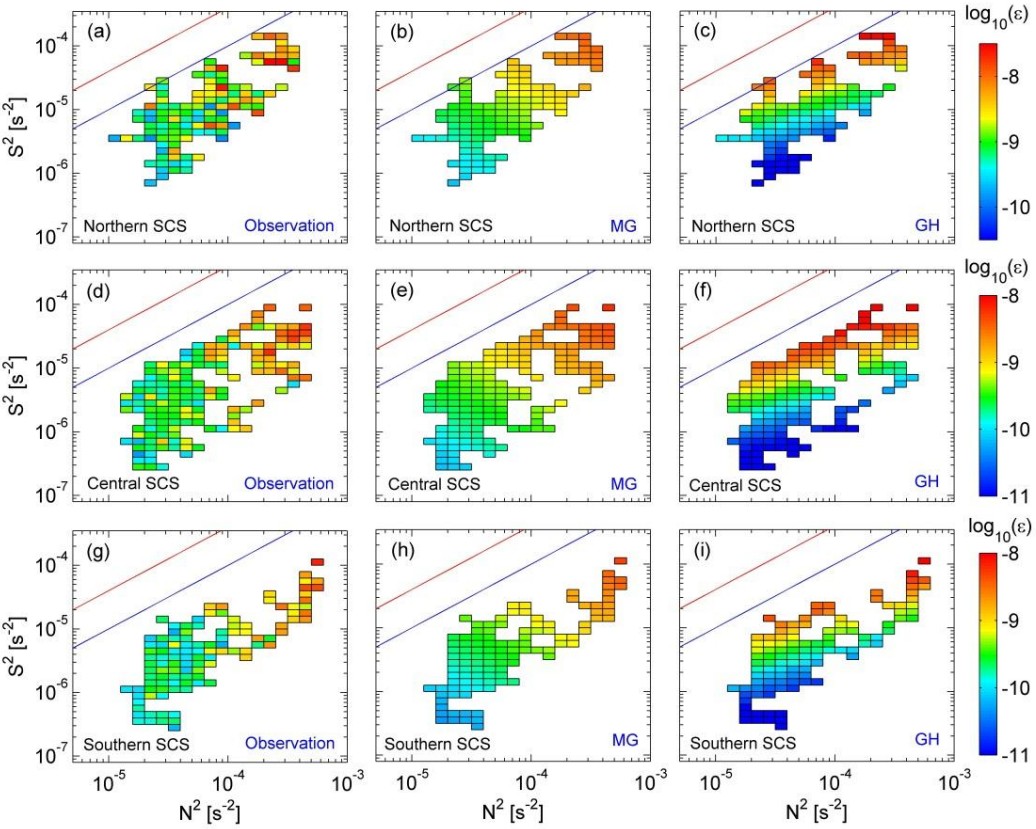

Fig. 8. Dissipation rates of observation $\varepsilon_{OB}$ (left column), MG model $\varepsilon_{MG}$ (middle column), and GH model $\varepsilon_{GH}$ (right column) averaged in bins of 16-m buoyancy frequency squared ($N^2$) and 16-m shear variance ($S^2$). All data affected by the

surface mixed layers or bottom mixed layers were excluded. (a-c) show the results of the stations in the northern SCS, (d-f) show the results of the stations in the central SCS, and (g-i) show the results of the stations in the southern SCS. The boundaries of $Ri$=0.25 (oblique red lines) and $Ri$=1 (oblique blue lines) are shown for reference.

Fig. 8 shows the distribution of dissipation rates (observed and modeled) in $N^2$ and $S^2$ space. The observed dissipation rates in the SCS (Fig. 8, left column) increase with increasing buoyancy frequency and shear. The GH model fails to reproduce these kinematic relationships (Fig. 8, right column). The dependence of $\varepsilon_{GH}$ on shear is too strong, with the dissipation rates underestimated in weak shear. $\varepsilon_{GH}$ also varies inversely with the buoyancy frequency for a given level of shear, contrary to the observation (Fig. 8, left column). Instead, the MG model dissipation rates (Fig. 8, middle column) display a pattern qualitatively consistent with the observed data (Fig. 8, left column). Both the observed and MG model dissipation rates scale positively with shear and the buoyancy frequency. In the northern SCS, the turbulence was more complicated than the predictions of the MG model. The MG model (Fig. 8b) underestimates the elevated dissipation rates that scattered in Fig. 8a; for example, the MG model underestimates the elevated dissipation rates at ($N^2$=6.5×10$^{-5}$ s$^{-2}$, $S^2$=5.0×10$^{-6}$ s$^{-2}$), ($N^2$=1.0×10$^{-4}$ s$^{-2}$, $S^2$=1.0×10$^{-5}$ s$^{-2}$), and ($N^2$=7.9×10$^{-5}$ s$^{-2}$, $S^2$=2.0×10$^{-5}$ s$^{-2}$).

Fig. 9 shows dissipation rate binned in terms of stratification or shear alone. They are equivalent to integrating the two-dimensional plots in Fig. 8 horizontally and vertically. Both models reproduce the slope of the dissipation rate versus the buoyancy frequency ($\varepsilon \propto N^2$) (Figs. 9a, 9c, and 9e), though the GH model dissipation rates are too large on average. However, the two models show large differences in the trend of the dissipation rate versus shear (Figs. 9b, 9d, and 9f). The MG model successfully captures the essential kinematic relationship between the dissipation rate and shear, whereas the GH model dissipation rates have a much steeper relationship with shear. Comparing the three regions, we find that the confidence intervals of the observed dissipation rates in the northern SCS (Figs. 9a and 9b) were wider than those in the central and southern SCS (Figs. 9c-9f). In addition, the observed dissipation rates in the northern SCS were slightly larger and showed greater fluctuations than the MG model dissipation rates (Figs. 9a and 9b). The wide confidence intervals and high observed dissipation rates in Figs. 9a and 9b mainly resulted from the elevated dissipation rates scattered in Fig. 8a. The MG model underestimated these elevated dissipation rates (comparing Fig. 8a with Fig. 8b). To explore these underestimations, we directly compared the model dissipation rates with the observed dissipation rates at three selected stations (Fig. 5, fourth column). For the stations from regions 3 and 4 (Figs. 5i and 5n), the relationships between the observed dissipation rates (pink curves) and the GH model dissipation rates (triangles) were poor, with the GH model dissipation rates deviating from the observed data by one order of magnitude. Instead, the MG model dissipation rates (stars) fared better than the GH model dissipation rates against the observed data. For station 6 from region 1 (Fig. 5d), the GH model dissipation rates also failed to overlap the observed data. Instead, the MG model dissipation rates agreed quite well with the observed data, except for the elevated dissipation rates induced by shear instability, for example, the MG model underestimated the elevated dissipation rates at depths of 175 to 195 m by more than one order of magnitude. The elevated dissipation rates scattered in Fig. 8a mainly resulted from the dissipation rates induced by shear instability. However, the GH

model dissipation rates seemed to agree with the elevated dissipation rates induced by shear instability (175-195 m). This agreement might be due to the fact that dissipation rates resulting from shear instability depend on the Richardson number and GH model also demonstrates Richardson number dependency.

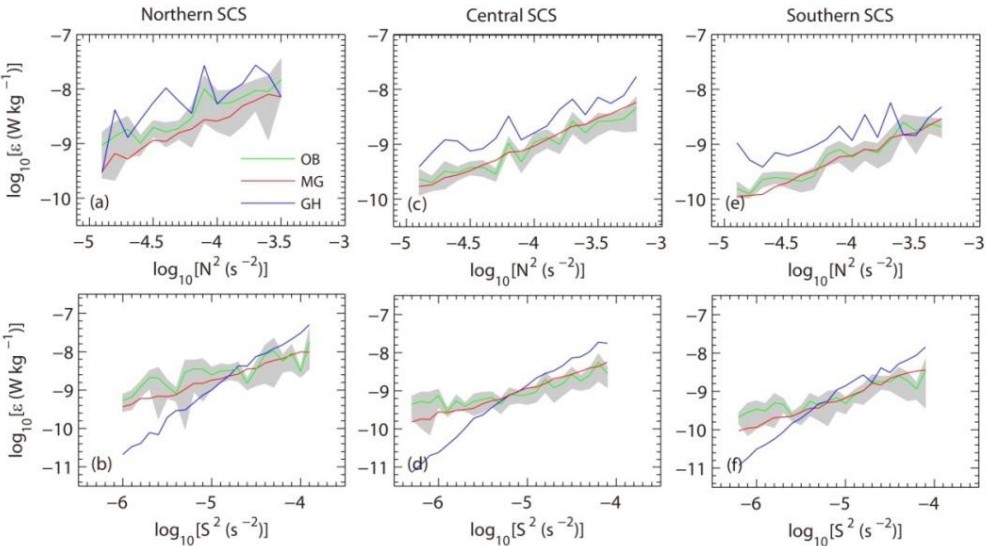

Fig. 9. Average dissipation rate calculated in bins of buoyancy frequency squared ($N^2$) and shear variance ($S^2$) for the northern SCS (a-b), central SCS (c-d), and southern SCS (e-f). The green, red, and blue curves are the results of the observation, MG model, and GH model, respectively. The grey shading indicates the 95% bootstrapped confidence interval for the observed dissipation rates.

To assess the efficacy of the models in estimating the dissipation rates, we show a direct comparison of observed dissipation rates vs. modeled dissipation rates in Fig. 10 and test the linear regression equation. It can be seen from Fig. 10 that the MG model predicts the magnitude of the dissipation rates much better than the GH model. For both models, the slopes of the linear fittings are smaller than one, which indicates that the models tend to overestimate the weak dissipation and underestimate the strong dissipation. Here we calculate the errors of the estimations to further assess the two models. The errors of the estimations are given by $|\log_{10}(\varepsilon_{OB}) - \log_{10}(\varepsilon_M)|$, where $\varepsilon_M$ represent the modeled dissipation rates. We define $\theta_M$ as the root mean square of $|\log_{10}(\varepsilon_{OB}) - \log_{10}(\varepsilon_M)|$. Thus the observed dissipation rates are mainly concentrated in the band of $[10^{-\theta_M} \ 10^{\theta_M}] \varepsilon_M$. For MG model, values of $\theta_{MG}$ are 0.59, 0.47, and 0.45 for the northern, central, and southern SCS, respectively. For GH model, values of $\theta_{GH}$ are 1.10, 1.01, and 0.97 for the northern, central, and southern SCS, respectively. Small values of $\theta_{MG}$ in the central and southern SCS indicate that MG model works better in the central and southern SCS than in the northern SCS. The MG model largely underestimates the elevated dissipation rates (Fig. 10a, $\varepsilon_{OB} > 2 \times 10^{-8}$ W kg$^{-1}$) in the northern SCS. These elevated dissipation rates correspond to the dissipation rates induced by shear instability. Values of $\theta_{MG}$ are smaller than that of $\theta_{GH}$ in all the regions, which indicate that errors of MG model are smaller

than those of GH model. For MG model, $\varepsilon_{OB}\sim$[0.26 3.89], [0.34 2.95], and [0.35 2.82] $\varepsilon_{MG}$ for the northern, central, and southern SCS, respectively, and for GH model, $\varepsilon_{OB}\sim$[0.08 12.59], [0.10 10.23], and [0.11 9.30] $\varepsilon_{GH}$ for the northern, central, and southern SCS, respectively.

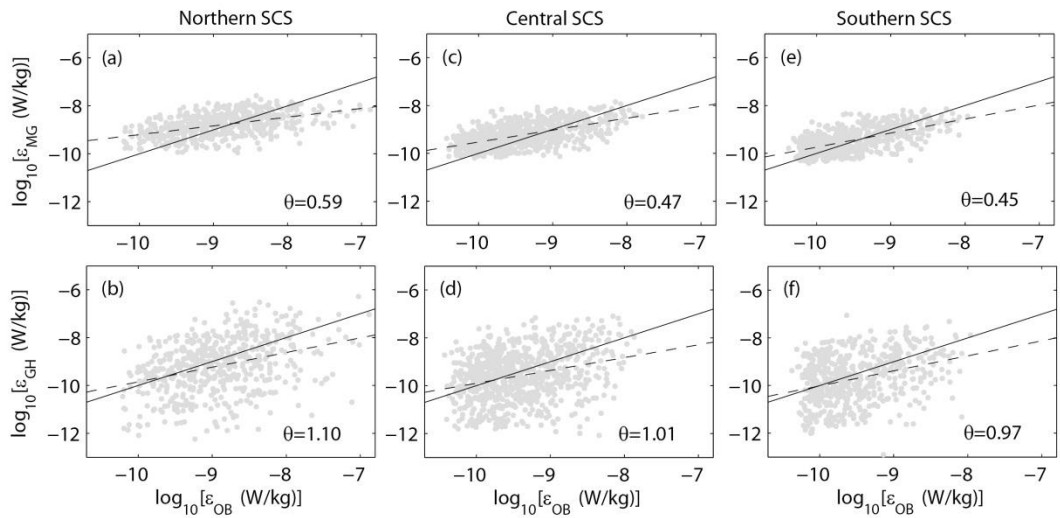

Fig. 10. Observed dissipation ($\varepsilon_{OB}$) plotted against modeled (top) MG and (bottom) GH dissipation for the northern SCS (a-b), central SCS (c-d), and southern SCS (e-f). The solid lines indicate the one-to-one relation: $\log_{10}(\varepsilon_{OB})=\log_{10}(\varepsilon_{GH})$ or $\log_{10}(\varepsilon_{OB})=\log_{10}(\varepsilon_{MG})$. The dash lines indicate the linear fittings of the data.

## 4 Discussion

Our observations indicate that turbulent mixing in the upper ocean of the SCS is spatially non-uniform, with strong turbulent mixing found in the northern SCS. This spatial pattern is consistent with the mixing distribution reported by Yang et al. [2016]. Our estimates of diapycnal diffusivity ($\sim10^{-5}\,\text{m}^2\,\text{s}^{-1}$) in region 1 are similar to those ($\sim10^{-5}\,\text{m}^2\,\text{s}^{-1}$) of Tian et al. [2009] but almost two orders of magnitude smaller than those ($\sim10^{-3}\,\text{m}^2\,\text{s}^{-1}$) reported by Yang et al. [2016]; these different values might be attributed to various factors such as estimation methods and observation seasons. Diapycnal diffusivities from Tian et al. [2009] and Yang et al. [2016] were estimated with parameterizations, which depends on reference dissipation. Different reference dissipations chosen in the parameterization can make the estimated diapycnal diffusivity different. In addition, the data used in the parameterization of Yang et al. [2016] span from 2005 to 2012 and cover all the year round while the microstructure data in our observation just cover one month. Seasonal and inter-annual variations of internal waves in the SCS [Huang et al., 2008; Yang et al., 2009] might affect the turbulent mixing.

GH model and MG model were derived from the eikonal model of Henyey et al. [1986] which is applicable to parameterize the dissipation controlled by wave-wave interactions that transfer energy from large-scale waves to small-scale waves [MacKinnon and Gregg, 2005]. The GH model is based on the assumption that the waves are statistically stationary,

with the energy of small-scale waves and the shear of the large-scale waves maintain a particular relationship through the Garrett-Munk (GM) spectrum [Garrett and Munk, 1975]. Its predication is typically evaluated for the internal wave field

with the GM spectral shape [Gregg, 1989]. The MG model is first proposed by MacKinnon and Gregg [2003] to parameterize the turbulence over the continental shelf. It is found to be suitable for the wave field of the continental shelf in which the energy and shear are dominated by the near-inertial motions, internal tides, or low frequency internal waves [MacKinnon and Gregg, 2003a; Palmer et al., 2008; van der Lee and Umlauf, 2011]. Recently it is found that MG model also successfully parameterizes the turbulent mixing in the upper ocean of deep sea [Xie et al., 2013].

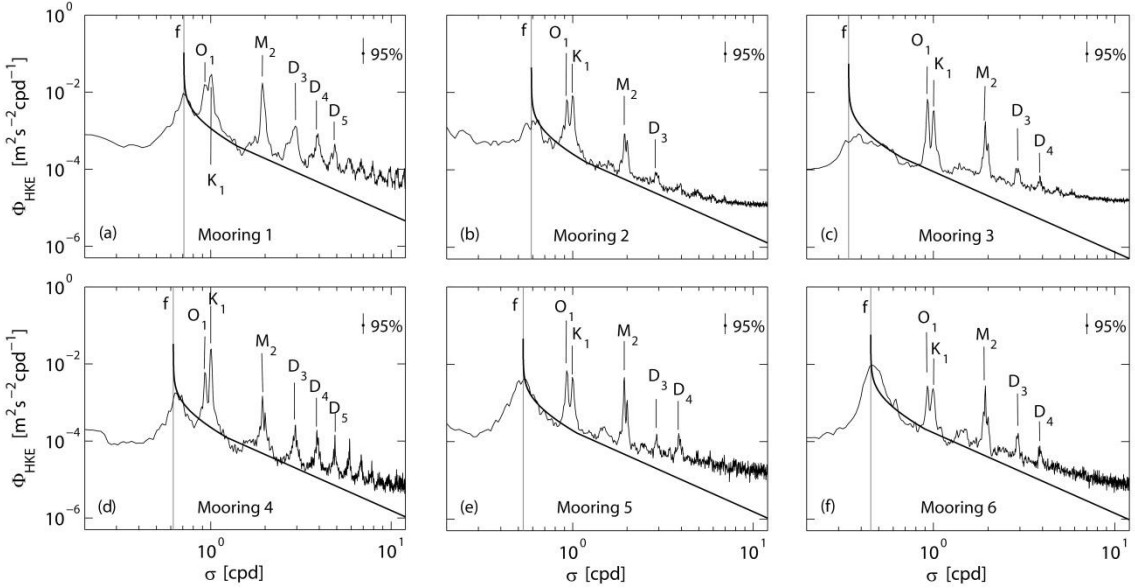

Fig. 11. (a-f) Rotary spectra (clockwise plus counterclockwise) of horizontal kinetic energy for the six moorings deployed in the SCS (1 cpd=1 day$^{-1}$). Spectra are averaged over $z \in [60:270]$ m. The canonical Garrett and Munk spectrum is shown for reference (smooth curve). The vertical lines represent various frequencies ($f$, $O_1$, $K_1$ …). The 95% statistical significance level is indicated by the vertical bar in the upper-right corner.

Statistical analysis shows the dissipation rates in the SCS to be proportional to both the shear and buoyancy frequencies, in marked contrast to the predictions of the GH model, but consistent with the predictions of the MG model. The disagreement of the GH model might be associated with the wave field in the SCS. Previous studies [Polzin et al., 1995; Wijesekera et al., 1993] have indicated that the predictions of the GH model would exhibit departure from the observed dissipation by more than one order of magnitude in regions where the wave field deviates from the GM spectrum. Thus, it is

appropriate to examine the wave field in the SCS. Data obtained from six moorings deployed in the SCS (Fig. 1a, yellow squares) were used to estimate the horizontal kinetic energy spectra. Though the data were obtained from different periods, they reflected the main characteristics of the wave field in the SCS. The spectra (Fig. 11) show significant peaks in the local inertial ($f$) and tidal frequencies (diurnal $O_1$ and $K_1$; semidiurnal $M_2$); these peaks imply that energy was primarily dominated

by the near-inertial motions and internal tides. Within the internal wave band, significant peaks were also observed at higher tidal harmonic frequencies such as $D_3$, $D_4$, and $D_5$ (respectively about 3, 4, 5 cycles per day). These higher tidal harmonic frequencies mainly result from nonlinear interaction between internal waves [van Haren, 2003; van Haren et al., 2002; Xie et al., 2010]. These energetic internal tides and harmonic internal waves cannot be well described by the GM spectrum. Furthermore, the spectra deviated from the GM spectrum at high frequencies ($\sigma > 3$ cpd), which is especially evident in the spectra of the moorings from the northern SCS (mooring 1) and southern SCS (mooring 3). These observations are not supportive of the assumption that the GH model is based on. In contrast, some of our observations support the MG model, such as the wave field is dominated by near-inertial waves and internal tides, and the dissipation rates scale positively with shear and stratification. Overall, the MG model succeeds in parameterizing the turbulence in the SCS, except for some elevated dissipation rates induced by shear instability. The MG model tends to underestimate these elevated dissipation rates. This is not surprising because the MG model, which is based on wave-wave interactions, represents bulk averages of turbulent properties and does not reproduce individual shear instability events [MacKinnon and Gregg, 2005].

## 5 Summary

We analyzed observations of turbulent dissipation and mixing in the SCS with microstructure data obtained from April 26 to May 23 2010. The observations are divided into four regions: region 1 is located to the west of the Luzon Strait, region 2 is located to the northeast of Hainan Island, region 3 is located in the central SCS, and region 4 is located in the southern SCS. Strong turbulent mixing was observed in region 1 with the mean $\langle\varepsilon\rangle_T$ reaching $1.8\times10^{-8}$ W kg$^{-1}$, which is 9 times and 12 times larger than the values in the central ($2.1\times10^{-9}$ W kg$^{-1}$) and southern ($1.5\times10^{-9}$ W kg$^{-1}$) SCS, respectively. Elevated $\langle\kappa\rangle_T$ were also found in region 1, i.e., $O$ ($10^{-5}$ m$^2$ s$^{-1}$), which is almost an order of magnitude higher than the values of central and southern SCS. The turbulent mixing in different regions displays different mixing features, to which shear variance and stratification have made significant contribution. In region 1, the shear was stronger and the stratification was weaker than those in other regions. Shear instability events occasionally occurred in these conditions and produced elevated dissipation and diapycnal diffusivity. Although the turbulent patches induced by shear instability were occasional and sparse, they significantly contributed to the turbulent mixing in the water column. In the central and southern SCS (region 3 and region 4), the water column was characterized by weak shear and strong stratification. Shear was no longer sufficient to produce subcritical Richardson numbers and the turbulence was weak. The strong spatial correlation between high dissipation rates and strong shear presented in the thermocline in region 1 suggests that shear was one of the important drivers of the elevated turbulent mixing. The analysis of surface winds, internal waves, and barotropic tides indicates that the spatial distribution of turbulent mixing with elevated dissipation rates and diapycnal diffusivity concentrated in region 1 does not result from the measure time or surface winds. The energetic internal waves and internal tides generated near the Luzon Strait are expected to make a dominant contribution to create this mixing pattern. Unfortunately, we have only one profile of microstructure measurement and short time-series of current velocity obtained by the shipboard ADCP for each station, thus it is impossible

to separate the internal waves of various frequencies and explore their respective contributions to the dissipation. In order to resolve the internal waves in various frequencies, a long time-series of fine-scale current velocities is required. We suggest further observations be done with frequent microstructure measurements and long time-series of current velocity measurements to identify the dominant mixing mechanism in the northern SCS.

To predict realistic climate and circulation, mixing must be accurately represented in ocean models. Mapping of the dissipation rates throughout the ocean is a daunting task. However, this task can be made considerably easier if mixing can be estimated from more easily observed or modeled quantities, such as shear, stratification, and latitude. Two models (the GH model and MG model) were evaluated for parameterizing the dissipation rate in the SCS. Statistical analysis shows the dissipation in the SCS to be proportional to both the shear and buoyancy frequencies, in marked contrast to the predictions of

the GH model, but consistent with the predictions of the MG model. The replication of the turbulence behavior greatly depends on the correct choice of model and appropriate tuning of the free parameters. The resolution of the shear and stratification is another factor in determining the success of models in parameterizing the turbulence [Palmer et al., 2013]. Although the MG model can reproduce the dissipation in the SCS for our chosen vertical resolution (16 m), whether the distribution of the observed dissipation would change with finer resolution of shear and stratification is still an open problem.

However, at least on the scale of internal waves (16 m), the MG model is clearly a better model than the GH model for the parameterization of turbulence in the upper ocean of the SCS, which provides a useful reference for modelers. Additional data with higher resolution are required to robustly fix this model in the near future.

**Author contribution**

Xiao-Dong Shang and Gui-Ying Chen designed and carried out the experiments. Chang-Rong Liang prepared the

manuscript with contributions from all co-authors.

**Competing interests**

The authors declare that they have no conflict of interest.

**Acknowledgments**

This work is supported by the National Natural Science Foundation of China: 41630970, 41376022, 41676022, and

41521005. The data we used is from South China Sea Institute of Oceanology.

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
