# Peer review of "Spatial distribution of turbulent mixing in the upper ocean of the South China Sea"

_Ocean Science, 2016_

## Referee Comment (RC1) · H. Burchard (Referee) · 10 Feb 2017

Review of

**Spatial distribution of turbulent mixing in the upper ocean of the South China Sea**

by

**Xiao-Dong Shang, Chang-Rong Liang and Gui-Ying Chen**

submitted to

**Ocean Science**

**General comments**

This is an interesting manuscript presenting a new data set observed in the South China Sea (SCS) by means of micro-structure shear (MSS) profilers. In four different geographical regions of the SCS, a total of 82 MSS profiles were obtained, covering the first 500m of the water column. Analyses mainly in terms of the dissipation rate and the diapycnal diffusivity are performed and discussed in comparison to the buoyancy and shear frequencies squared and the gradient Richardson number. Comparison of the results for the dissipation rate to the dissipation rate parameterisations by Gregg (1989) and to the parameterisation by MacKinnon and Gregg (2005) is performed. In most situations, the latter model better represents the data. To explain this, internal wave spectra are derived from observations of 5 moorings in the SCS. Considerable deviations from the Garrett-Munk spectra, on which the Gregg (1989) model is based, explain the weakness of this model.

Having said this, the manuscript is generally publishable in Ocean Science. However, I have some concerns which need to be considered by the authors before acceptance can be recommended. Major revisions are required.

One concern is the lack of physical interpretation of the results. The authors should explain why certain areas show large or small shear and stratification, respectively. The role of high-amplitude internal waves entering from the Luzon Strait and their effect on mixing in the SCS needs to be discussed.

The study needs a better motivation. Which is the major knowledge gap to be filled? This should come out as a result from the introduction. In the moment it reads a bit like a report to present new data for the first time.

The shear estimated from 16m-bins is very coarsely resolved. Therefore, the gradient Richardson number calculated on that shear might be substantially underestimated. This needs to be discussed in more depth (not only in section 3.3).

**Specific comments**

Line 24: wrong unit (should be m2/s)

Lines 37/38: "large numbers of … tides": better expression needed

Lines 53/54: not clear why these parameterisations are important for ocean models. Please explain, how those could be used, since I am not aware of an ocean model using these parameterisations. See also line 310, where something similar is postulated.

Lines 54-56: here, a better motivation is needed.

Line 58: what does "LT" stand for?

Line 81: What is the detection limit for the TurboMAP profiler. You measure here very low dissipation rates of $10^{-10}$ W/kg. Are they still above the limit?

Line 141: here the gradient Richardson number is defined. How is it calculated? Already here, and not as late as in section 3.3, you should discuss the consequences of a very low resolution shear estimate. Do also refer to the literature, how others cope with such low resolution of the shear when calculating Ri. I assume that at many other locations in your observations Ri<1/4 should occur (otherwise the disspation rate would be lower), but you do not resolved it.

Line 151: So, why is there strong shear and weak stratification in region 1 and vice versa in the other regions. Is it external and/or internal tides which are different across the SCS? Is it different wind regimes? In general, we need more physical oceanography here.

Line 168: This overturn should have gone together with locally increased shear which is not resolved in the observations.

Line 176. There is some confusion about the background value for eddy diffusivity in the ocean. In line 25, it is $10^{-5}$, in line 170, it is $5 \times 10^{-6}$, and here it is of the order of $10^{-6}$. These are considerably different values. Please clarify.

Line 181: What is the physical meaning of depth and time averaged eddy diffusivity? Eddy diffusivity is a ratio (between flux and gradient), and the average of a ratio does not much sense to me. What is the additional information it gives in addition to the averaged dissipation rate (which makes sense)?

Line 196: Does also the tidal phase at which the observations where taken matter? If not, why not? What about the wind forcing? Does it vary, does it matter? See Burchard & Rippeth (2009), where wind-induced shear across the thermocline matters.

Burchard, H., and T.P. Rippeth, 2009: Generation of bulk shear spikes in shallow stratified tidal seas, *J. Phys. Oceanogr.*, 39, 969-985.

Fig. 6b: wrong unit for eddy diffusivity.

Equation after line 222: Something is wrong with the dimensions here. When f is 1/s, then $1.8 \times 10^{-6}$ should be m2/s2. Replace $1.8 \times 10^{-6}$ with a variable name and explain amount and unit in the text. Also, some of the brackets seem to denote an argument for a function $\cosh^{-1}$ and some denote a factor. Please clarify.

Line 222: Express cph also in Si units (1/s). Sometime cph and sometimes cpd is used, which I find confusing.

The two parametersations GH and MG should be explained for their physical reasoning. They are for different environments, deep ocean (GH) and shelf sea (MG), as I understand.

Lines 234/235: Are these data also for the thermocline region, or is it over the entire water column except for boundary layers?

Line 237: Here, a method for calculating Ri is explained. Is it different than before? Give this explanation at the first occurrence of Ri.

Fig. 7: Add locations (regions) to the plot.

Line 276: What is "fared"?

Line 298: These different techniques and seasons should be discussed with respect to their effect on observed dissipation rates.

Line 314: I had to look up the word eikonal. And it would be good, if the authors could briefly explain the eikonal model.

Line 327: typo "flied".

Line 330: Add "tidal" in front of "frequencies".

Line 332:What are the D3, D4 and D5 frequencies?

Line 335/336: Sentence is a repetition of what has been written further up.

Line 340/341: How can a model for bulk averages be used for constructing profiles (such as in fig. 5)?

Line 345: word missing after "observed".

---

## Referee Comment (RC2) · Anonymous Referee #2 · 17 Feb 2017

The authors conducted an extensive field campaign to survey turbulence in the South China Sea and reported a spatial pattern of turbulent intensity from the observed data. They also compared the observed data against two theoretical models. The results are well known. I found no new information. I appreciate that the amount of work involved in the data collection, but as far as the science concerns the present manuscript reads like a data report. I found no new scientific finding and no new facts other than the survey was conducted in the South China Sea. Unless Ocean Science accept a manuscript aimed at a data report, I would not recommend this manuscript for an official scientific paper. For their revision purpose I will comment on this manuscript as followed:

1. Material and methods should be explained in more detail. Also those mooring data outside the observation period should be removed from the text. The Luzon Strait is not well known to the audience. Indicate where LS is.

2. P.4 line83: "caused by instrument vibrations" If so, you can verify the vibration with accelerometer data. Those are mostly electronic noise.

3. You have to focus the science. What is deriving high turbulence in Region 1. Most likely internal tides are playing a major role in generation of turbulence.

4. Discussion should be separated from Summary. The summary should summarize both results and a punch line of the discussion.
* * *

---

## Author Response (AR1)

Our responses to Reviewer 1:
Note: The reviewer's original comments are in black, and our responses are in blue.

**General comments**

This is an interesting manuscript presenting a new data set observed in the South China Sea (SCS) by means of micro-structure shear (MSS) profilers. In four different geographical regions of the SCS, a total of 82 MSS profiles were obtained, covering the first 500m of the water column. Analyses mainly in terms of the dissipation rate and the diapycnal diffusivity are performed and discussed in comparison to the buoyancy and shear frequencies squared and the gradient

Richardson number. Comparison of the results for the dissipation rate to the dissipation rate parameterizations by Gregg (1989) and to the parameterization by MacKinnon and Gregg (2005) is performed. In most situations, the latter model better represents the data. To explain this, internal wave spectra are derived from observations of 5 moorings in the SCS. Considerable deviations from the Garrett-Munk spectra, on which the Gregg (1989) model is based, explain the weakness of this model.

Having said this, the manuscript is generally publishable in Ocean Science. However, I have some concerns which need to be considered by the authors before acceptance can be recommended. Major revisions are required.

One concern is the lack of physical interpretation of the results. The authors should explain why certain areas show large or small shear and stratification, respectively. The role of high-amplitude internal waves entering from the Luzon Strait and their effect on mixing in the SCS needs to be discussed.

Responses:
Thank you for your good advice. There are a large amount of internal waves (tides) in the South China Sea, which has been reported in many literatures, such as Niwa and Hibiya (2004), Zhao et al. (2004), Klymak et al. (2006), Jan et al. (2007). Most of internal waves originate in Luzon Strait and propagate northwestwards through the deep water zone near Luzon Strait to the continental shelf (Alford et al. 2015). Mooring data (Lien et al. 2014) indicate that these internal waves would induce strong shear.
A comparison of the spatial distributions of turbulent mixing and internal waves indicated that the internal waves are expected to make a dominant contribution to elevate the turbulent mixing in west of the Luzon Strait. For reasons why strong shear and elevated dissipation occurred in west of the Luzon Strait, we have strengthened the discussion in the revised text, please see more detail in lines 246-275 and 422-431.

The study needs a better motivation. Which is the major knowledge gap to be filled? This should come out as a result from the introduction. In the moment it reads a bit like a report to present new data for the first time.
Responses:
We thank the reviewer for this good suggestion. There are two motivations for this study: firstly exploring the mixing features and mixing regimes in different regions of the SCS and secondly assessing two parameterizations with the microstructure data.

Many microstructure measurements have been conducted in the SCS. There is no doubt that these measurements have greatly aided our knowledge of turbulent mixing in the SCS. However, the microstructure measurements are localized and scattered with most of them focusing on the northern SCS. The mixing features and mixing regimes in different regions of the SCS are still not fully understood. With the microstructure data in 2010, we present the spatial distribution of turbulent mixing in the upper ocean of the SCS and explore the mixing features and mixing regimes in different regions of the SCS in great detail. In the revised text, we strengthened the discussion on energy sources for the turbulent mixing (lines 246-275 and 422-431). Our observation indicated that strong turbulent mixing mainly occurred in west of the Luzon Strait where there are strong shear and weak stratification, and internal waves made a dominant contribution to the elevated turbulent mixing in west of the Luzon Strait.

Another motivation for this paper is the assessment of two parameterizations (GH and MG models). Though many microstructure measurements have been conducted in the SCS, none of the two models has been assessed against the dissipation in the SCS. It remains unknown which parameterization can successfully reproduce the dissipation in the SCS and why. In manuscript, we assess the two parameterizations with the dissipation data of the SCS, which would provide useful tools for ocean researchers. In fact, the microstructure measurements in the ocean are much fewer and more difficult than fine-structure measurements (i.e., CTD and ADCP measurements) in the ocean, especially in the deep sea. Thus to understand the spatial distribution and seasonal variation of the turbulent mixing in the ocean, researchers often turn to the parameterizations (Wu et al. 2011; Jing and Wu 2010). The assessment of parameterizations in the SCS would provide reference for researchers on the selection of parameterization to study the turbulent mixing in the SCS. The assessment of parameterizations can also provide reference for modelers. Sea models have success in reproducing the water column structure in seasonally stratified shelf seas (Holt and Umlauf 2008; Simpson and Bowers 1981; Simpson and Hunter 1974). However, models need to calibrate a background mixing level for the correctly prediction (Rippeth 2005). The requirement of calibration reduces the success of models on shelf-wide scales. Before the water column structure in shelf seas can be modeled realistically, the distribution of mixing must be established and the major mixing processes identified and parameterized. Parameterizations would provide a reference of the turbulence mixing for the modelers.

We have added the related content in the revised text as the reviewer suggested. Please see lines 52-74 in the revised text.

The shear estimated from 16m-bins is very coarsely resolved. Therefore, the gradient Richardson number calculated on that shear might be substantially underestimated. This needs to be discussed in more depth (not only in section 3.3).

Responses:

We understand the reviewer's concern. There is no doubt that the resolution of shear might affect the values of Richardson number. Unfortunately, we have only 16 m shear data, which prevents us from discussing the influence of shear resolution on the Richardson number. In our text,

Richardson number was estimated following MacKinnon and Gregg (2005), see their Fig. 5. In the previous literatures, different shear resolutions were used to calculate the Richardson number, which range from 2 m to 16 m (MacKinnon and Gregg 2003; 2005; van der Lee and Umlauf 2011; Xie et al. 2013; Yang et al. 2014). High resolution of shear (2-4 m) was often used on the shelf area to catch the small scale internal waves. Low resolution of shear (8-16 m) was often used in deep water due to the large depth of the water column. Our observations mainly located in deep water, so the ADCP vertical resolution was set in 16 m to cover more depth. Although the resolution of 16 m might miss some overturning in our observation, it does not affect the comparison of Richardson number in different regions too much. We have strengthened the discussion on this in the revised text, see lines 177-183.

**Specific comments**

Line 24: wrong unit (should be m2/s)

Responses:

Thanks for reminding. We have corrected the mistake, see line 26 in the revised text.

Lines 37/38: "large numbers of … tides": better expression needed

Responses:

We have changed "large numbers of … tides" into "numerous … tides", see line 40 in the revised text.

Lines 53/54: not clear why these parameterisations are important for ocean models. Please explain, how those could be used, since I am not aware of an ocean model using these parameterisations. See also line 310, where something similar is postulated.

Responses:

We apologize for this confusion due to our inaccurate statement in the original text. Parameterization is important mainly due to its reference to modelers. Shelf sea models have success in reproducing the water column structure in seasonally stratified shelf seas (Holt and Umlauf 2008; Simpson and Bowers 1981; Simpson and Hunter 1974). However, models need to calibrate a background mixing level for the correctly prediction (Rippeth 2005). The requirement of calibration reduces the success of models on shelf-wide scales. Before the water column structure in shelf seas can be modeled realistically, the distribution of mixing must be established and the major mixing processes identified and parameterized. Parameterizations would provide a reference of the turbulence mixing for the modelers. Of course, we hope that these parameterizations can be applied to model in the near future.

Lines 54-56: here, a better motivation is needed.

Responses:

We have strengthened this in the revised text, see lines 52-74.

Line 58: what does "LT" stand for?

Responses:

We apologize for this confusing abbreviation. "LT" stands for "local time''. We have changed "LT" into "local time'' in the revised text, see lines 76-77.

Line 81: What is the detection limit for the TurboMAP profiler. You measure here very low dissipation rates of $10^{-10}$ W/kg. Are they still above the limit?

Responses:

The noise level of the TurboMAP profiler is $\varepsilon \sim 10^{-10}$ W/kg (Matsuno and Wolk 2005; Wolk et al. 2002). TurboMAP profiler resolves dissipation rates as low as $5 \times 10^{-10}$ W/kg. Lower values of dissipation rates can be inferred by comparing the measured spectra against the assumed universal form. We have added this in the revised text, see line 117.

Line 141: here the gradient Richardson number is defined. How is it calculated? Already here, and not as late as in section 3.3, you should discuss the consequences of a very low resolution shear estimate. Do also refer to the literature, how others cope with such low resolution of the shear when calculating Ri. I assume that at many other locations in your observations Ri<1/4 should occur (otherwise the disspation rate would be lower), but you do not resolved it.
Responses:
Richardson number was estimated following MacKinnon and Gregg (2005), see their Fig. 5. Firstly calculate 16-m shear and 2-m buoyancy frequency, then interpolate the 2-m buoyancy frequency to the 16-m shear grids, and then calculate the Richardson number with the shear and buoyancy frequency. In the previous literatures, different shear resolutions were used to in the calculation of the Richardson number, which range from 2 m to 16 m (MacKinnon and Gregg 2003; 2005; van der Lee and Umlauf 2011; Xie et al. 2013; Yang et al. 2014). High resolution of shear was often chosen (2-4 m) on the shelf area to resolve the small scale internal waves. Low resolution of shear was often chosen (8-16 m) in deep water due to the large depth of the water column. Our observations mainly located in deep water, so the ADCP vertical resolution was set in 16 m to cover more depth.
We agree with the reviewer that 16-m resolution might miss some overturning in our observation. Unfortunately, we have only 16-m shear data, which prevents us from discussing the influence of shear resolution on the Richardson number. In spite of this, it does not affect the comparison of Richardson number in different regions too much since the same shear resolution was used in the SCS. We have strengthened the discussion on this in the revised text, see lines 177-183.

Line 151: So, why is there strong shear and weak stratification in region 1 and vice versa in the other regions. Is it external and/or internal tides which are different across the SCS? Is it different wind regimes? In general, we need more physical oceanography here.
Responses:
We thank the reviewer for this good question. There are a large amount of internal waves (tides) in the South China Sea, which has been reported in many literatures, such as Niwa and Hibiya (2004), Zhao et al. (2004), Klymak et al. (2006), Jan et al. (2007). Most of internal waves originate in Luzon Strait and propagate northwestwards through the deep water zone near Luzon Strait to the continental shelf (Alford et al. 2015). Mooring data and microstructure measurements (Laurent 2008; Lien et al. 2014) indicate that internal waves would induce strong shear and produce elevated turbulence.
A comparison of the spatial distributions of turbulent mixing, winds, and internal waves indicated that elevated turbulent mixing in west of Luzon Strait (region 1) does not result from the effect of surface winds. The internal waves are expected to make a dominant contribution to elevated turbulent mixing and shear in west of the Luzon Strait. Unfortunately, we have only one profile of microstructure measurement and short time series (about one hour) of current velocity obtained by the ADCP for each station, thus it is impossible to separate the internal waves in various frequencies and explore their respective contributions to the dissipation. We have strengthened the discussion on this in the revised text, see lines 246-275.

Line 168: This overturn should have gone together with locally increased shear which is not resolved in the observations.
Responses:
We agree with the reviewer. Small overturning might be missed in our observation due to the low shear resolution.

Line 176. There is some confusion about the background value for eddy diffusivity in the ocean. In line 25, it is $10^{-5}$, in line 170, it is $5 \times 10^{-6}$, and here it is of the order of $10^{-6}$. These are considerably different values. Please clarify.
Responses:

We apologize for this confusion. Diapycnal diffusivity from turbulent mixing in the open ocean thermocline ranges from $5 \times 10^{-6}$ to $3 \times 10^{-5}$ m$^2$/s (Gregg 1998; Polzin et al. 1995). We have clarified this in the revised text, see lines 26-27, 210, and 215-217.

Line 181: What is the physical meaning of depth and time averaged eddy diffusivity? Eddy
diffusivity is a ratio (between flux and gradient), and the average of a ratio does not much sense to me. What is the additional information it gives in addition to the averaged dissipation rate (which makes sense)?
Responses:
In steady state, dissipation should equal the rate of transfer from the internal waves to turbulence mixing. Column integrated dissipation $\int_{-H}^{0} \rho \varepsilon dz$ (W/m$^2$) represents the rate of energy dissipation per square meter, which is often used to calculate the energy of internal waves losing to dissipation. Here, in order to compare the magnitude of dissipation and diffusivity in different stations and regions, we use the averaged dissipation rate, which can remove the influence of different thermocline depths on the comparison. Averaged eddy diffusivity and dissipation rate are also used
to discuss the influence of wind and internal waves on the distribution of turbulence mixing. In addition, averaged eddy diffusivity is used to compare with the values in the open ocean.

Line 196: Does also the tidal phase at which the observations where taken matter? If not, why not? What about the wind forcing? Does it vary, does it matter? See Burchard & Rippeth (2009), where
wind-induced shear across the thermocline matters. Burchard, H., and T.P. Rippeth, 2009: Generation of bulk shear spikes in shallow stratified tidal seas, J. Phys. Oceanogr., 39, 969-985.
Responses:
The barotropic tides extracted at location of (18$^o$N, 114$^o$E) were used in the discussion without considering the tidal phase. The tidal phase does not affect our discussion too much because the
bias in the arrival of spring-neap tides in different locations of the SCS is small. Fig A1 shows the time series of the barotropic tidal velocity predicted from TPXO 7.1 for three locations: (21$^o$N, 119$^o$E), (18$^o$N, 114$^o$E), and (10$^o$N, 114$^o$E). One can see from Fig A1 that bias in the arrival of spring-neap tides between northern location (21$^o$N, 119$^o$E) and southern location (10$^o$N, 114$^o$E) was less than 3 hour. Related content has been added in the revised text, see lines 237-239.

We have added discussion on the effect of surface winds on turbulent mixing and shear in the revised text, see lines 246-265.

[Figure]

Fig. A1. Time series of the barotropic tidal velocity predicted from TPXO 7.1 for three locations: $(21^\circ N, 119^\circ E)$, $(18^\circ N, 114^\circ E)$, and $(10^\circ N, 114^\circ E)$.

Fig. 6b: wrong unit for eddy diffusivity. Equation after line 222: Something is wrong with the dimensions here. When f is 1/s, then $1.8 \times 10^{-6}$ should be $m^2/s^2$. Replace $1.8 \times 10^{-6}$ with a variable name and explain amount and unit in the text. Also, some of the brackets seem to denote an argument for a function $\cosh^{-1}$ and some denote a factor. Please clarify.

Responses:
Thanks for reminding. We have corrected the unit in Fig. 6. $\cosh^{-1}$ denotes inverse hyperbolic cosine function not a factor. The unit for $1.8 \times 10^{-6}$ is $m^2/s^2$. Actually a reference Coriolis frequency $f_0$ has been included in $1.8 \times 10^{-6}$. We have revised the manuscript according to the reviewers.

Line 222: Express cph also in Si units (1/s). Sometime cph and sometimes cpd is used, which I find confusing. The two parameterizations GH and MG should be explained for their physical reasoning. They are for different environments, deep ocean (GH) and shelf sea (MG), as I understand.
Responses:
We apologize for this confusion. $1 \text{ cph} = 1.7 \times 10^{-3} \text{ s}^{-1}$ and $1 \text{ cpd} = 1 \text{ day}^{-1} = 1.1574 \times 10^{-5} \text{ s}^{-1}$. To avoid confusion, we have changed the unit "cph" into "$s^{-1}$" and added "(1 cpd=1 day$^{-1}$)" in the revised text. Some of physical reasoning of the two parameterizations has been summarized in lines 358-367. For more information about the two parameterizations, one can refer to (MacKinnon and Gregg, 2003a). We are sorry for that we can't describe these two parameterizations better than
them.

Lines 234/235: Are these data also for the thermocline region, or is it over the entire water column except for boundary layers?
Responses:
It is over the entire water column except for boundary layers.

Line 237: Here, a method for calculating Ri is explained. Is it different than before? Give this explanation at the first occurrence of Ri.
Responses:
The difference between the two is that 2-m buoyancy frequency was used at the first occurrence of Ri and 16-m buoyancy frequency was used at latter. We have clarified this in the revised text, see

Fig. 7: Add locations (regions) to the plot.

Responses:

Thanks for reminding. Locations (regions) have been added to the plot.

Line 276: What is "fared"?

Responses:

"fared" means ''show''

Line 298: These different techniques and seasons should be discussed with respect to their effect on observed dissipation rates.

Responses:

We thank the reviewer for this good suggestion. We have strengthened the discussion on this in the revised text, see lines 408-413.

Line 314: I had to look up the word eikonal. And it would be good, if the authors could briefly explain the eikonal model.

Responses:

Henyey et al. (1986) construct the analytic model by equating $\varepsilon$ to the net flux of energy passing out of the internal wave spectrum at large wave number, which they take as $2k_3^c$, corresponding to a 5-m vertical wavelength. Because the energy flows toward both lower and higher wave numbers,

$$\varepsilon_{HWF} = \frac{1-r}{1+r} \int_f^N \Phi_{flux}(2k_3^c,\omega)\,d\omega, \qquad (1)$$

where $\Phi_{flux}(k_3,\omega)$ is the energy flux spectrum and $r$ is the ratio of the flux passing $2k_3^c$ toward lower wave numbers to the flux going toward higher wave numbers. The flux spectrum is formulated using ray-tracing equations to describe waves having high wave numbers propagating through a field in which most of the energy resides at low wave numbers. In keeping with a ray-tracing approach, $\Phi_{flux}(k_3,\omega)$ can be expressed as the product of the GM model energy spectrum $\Phi_E(k_3,\omega)$ and the rate at which the vertical wave number changes as the packet passes through the spectrum,

$$\Phi_{flux}(k_3,\omega) = \Phi_E(k_3,\omega) \left|\frac{dk_3(\omega)}{dt}\right| \text{ and} \qquad (2)$$

$$\Phi_E(k_3,\omega) = \frac{2b^3\beta_*NN_0E_{GM}}{\pi(\beta_*+\beta)^2} \frac{f}{\omega(\omega^2-f^2)^{1/2}}, \qquad (3)$$

where $\beta_* = j_*\pi N/N_0$ and $\beta = bk_3$. Because the flux is evaluated at a wave number in the roll-off region of the energy spectrum,

$$\Phi_E(k_3,\omega) = \frac{k_3}{k_3^c}\Phi_E(k_3^c,\omega) \qquad k_3 > k_3^c, \qquad (4)$$

where $k_3^c = (3Ri_c j_* bE_{GM})^{-1}$, For large wave numbers, the ray-tracing equations give

$$\left|\frac{dk_3}{dt}\right| = |\boldsymbol{S} \cdot \boldsymbol{k}|, \qquad (5)$$

where $\boldsymbol{S}$ is the shear vector. If $\boldsymbol{S}$ and $\boldsymbol{k}$ are uncorrelated,

$$\left|\frac{dk_3}{dt}\right| = Nk_h \left[\frac{1}{2}\langle Ri^{-1}\rangle\right]^{1/2}. \qquad (6)$$

where $k_h = k_3[(\omega^2 - f^2)/(N^2 - \omega^2)]^{1/2}$. In evaluating $\langle Ri^{-1} \rangle$, they take account of the additional shear contributed by wave numbers between $k_3^c$ and $2k_3^c$,

$$\langle Ri^{-1} \rangle = Ri_c^{-1}[1 + \ln(k_3/k_3^c)]. \quad (7)$$

Using (2)-(7) in (1) leaves the integral

$$\int_f^N \omega^{-1}(N^2 - \omega^2)^{-1/2}\, d\omega = N^{-1}\cosh^{-1}(N/f)$$

Following Munk (1981) $Ri_c^{-1}$=0.5, $k_3/k_3^c = 2$, and r =0.4,

$$\varepsilon_{HWF} = 0.33 f^{-1}[4\pi^{-1}j_*bE_{GM}f]^2 \cosh^{-1}(N/f).$$

Line 327: typo "flied".

Responses:

Thanks for reminding. We have corrected the typo "flied".

Line 330: Add "tidal" in front of "frequencies".

Responses:

Thanks for reminding. We have added the missing word "tidal".

Line 332:What are the $D_3$, $D_4$ and $D_5$ frequencies?

Responses:

$D_3$, $D_4$ and $D_5$ are the higher tidal harmonic frequencies, i.e., $D_3$=$D_1$+$D_2$, $D_4$=$D_2$+$D_2$, and $D_5$= $D_2$+$D_3$, where $D_1$ and $D_2$ represent the diurnal and semidiurnal tidal frequencies, respectively. These higher tidal harmonic frequencies mainly result from nonlinear interaction between internal waves [van Haren, 2002; van Haren, 2003; Xie, 2010]. We have cited the work of van Haren (2003), van Haren et al. (2002), and Xie et al. (2010) in the revised text, see lines 377-379.

Line 335/336: Sentence is a repetition of what has been written further up.

Responses:

Thanks for reminding. We have deleted repetition "The GH model is typically evaluated for the wave field with the GM spectral shape".

Line 340/341: How can a model for bulk averages be used for constructing profiles (such as in fig.5)?

Responses:

With the calculated shear $S(z)$ and buoyancy frequency $N(z)$, profiles can be constructed from the equations of $\varepsilon_{MG}(z) = \varepsilon_0 \frac{N(z)}{N_0} \frac{S(z)}{S_0}$ and $\varepsilon_{GH} = 1.8 \times 10^{-6} \left[ f\cosh^{-1}\left(\frac{N_0}{f}\right) \right] \left[ \frac{S(z)^4}{S_{GM}^4} \right] \left[ \frac{N(z)^2}{N_0^2} \right]$.

Line 345: word missing after "observed".

Responses:

Thanks for reminding. We have added the missing word "dissipation".

The authors conducted an extensive field campaign to survey turbulence in the South China Sea and reported a spatial pattern of turbulent intensity from the observed data. They also compared the observed data against two theoretical models. The results are well known. I found no new information. I appreciate that the amount of work involved in the data collection, but as far as the science concerns the present manuscript reads like a data report. I found no new scientific finding and no new facts other than the survey was conducted in the South China Sea. Unless Ocean Science accept a manuscript aimed at a data report, I would not recommend this manuscript for an official scientific paper. For their revision purpose I will comment on this manuscript as followed:

Responses:

We thank the reviewer for the comment. In this paper, the turbulent mixing in the South China Sea (SCS) is analyzed from two aspects, not just a simple data report. Firstly we explore the mixing features and mixing regimes in the SCS in great detail. Secondly we assess two parameterizations with the microstructure data.

Many microstructure measurements have been conducted in the SCS. There is no doubt that these measurements have greatly aided our knowledge of turbulent mixing in the SCS. However, the microstructure measurements are localized and scattered with most of them focusing on the northern SCS. The mixing features and mixing regimes in different regions of the SCS are still not fully understood. With the microstructure data in 2010, we present the spatial distribution of turbulent mixing in the upper ocean of the SCS and explore the mixing features and mixing regimes in different regions of the SCS. In the revised text, we strengthened the discussion on energy sources for the turbulent mixing (lines 246-275 and 422-431). Our observation indicated that strong turbulent mixing mainly occurred in west of the Luzon Strait where there are strong shear and weak stratification, and internal waves made a dominant contribution to the elevated turbulent mixing in west of the Luzon Strait.

Another work in this paper is the assessment of two parameterizations (GH and MG models).

Though many microstructure measurements have been conducted in the SCS, none of the two models has been assessed against the dissipation in the SCS. It remains unknown which parameterization can successfully reproduce the dissipation in the SCS and why. In manuscript, we assess the two parameterizations with the dissipation data of the SCS, which would provide useful tools for ocean researchers. In fact, the microstructure measurements in the ocean are much fewer and more difficult than fine-structure measurements (i.e., CTD and ADCP measurements) in the ocean, especially in the deep sea. Thus to understand the spatial distribution and seasonal variation of the turbulent mixing in the ocean, researchers often turn to the parameterizations (Wu et al. 2011; Jing and Wu 2010). The assessment of parameterizations can also provide reference for modelers. Models often need to calibrate a background mixing level to correctly predict the physical phenomenon (Rippeth 2005). The requirement of calibration reduces the success of models on large scales since differing forcing mechanisms and mixing processes require specific methods and levels of tuning. Before the physical phenomenon can be modeled realistically, the distribution of mixing must be established and the major mixing processes parameterized. Parameterizations can provide a reference of the background mixing for the modelers.

According to reviewers' comments, we have strengthened the content of motivation (lines 52-74)

and added discussion on the effect of surface winds and internal waves on turbulent mixing and shear (lines 246-275) in the revised text.

1. Material and methods should be explained in more detail. Also those mooring data outside the observation period should be removed from the text. The Luzon Strait is not well known to the audience. Indicate where LS is.
Responses:
We thank the reviewer for this suggestion. Mooring data were used to show the feature of wave field in the SCS. We think that it should be kept in the text. We have strengthened the description of the material and methods in the revised text (lines 76-123). The location of Luzon Strait has been indicated in Fig .1.

2. P.4 line83: "caused by instrument vibrations" If so, you can verify the vibration with accelerometer data. Those are mostly electronic noise.
Responses:
We apologize for this confusion due to our inaccurate statement in the original text. They are vibration noise caused by the strumming of the suspension wires in the flow (Wolk et al. 2002). We have clarified in the revised text, see lines 108-109.

3. You have to focus the science. What is deriving high turbulence in Region 1. Most likely internal tides are playing a major role in generation of turbulence.
Responses:
We thank the reviewer for this good question. There are a large amount of internal waves (tides) in the SCS, which has been reported in many literatures, such as Niwa and Hibiya (2004), Zhao et al. (2004), Klymak et al. (2006), Jan et al. (2007). Most of internal waves originate in the Luzon Strait and propagate northwestwards through the deep water zone near Luzon Strait to the continental shelf (Alford et al. 2015). Mooring data (Lien et al. 2014) indicate that internal waves would induce strong shear. A comparison of the spatial distributions of turbulent mixing, winds, and internal waves suggests that the elevated turbulent mixing and shear in west of the Luzon Strait (region 1) does not result from the effect of surface winds. The internal waves are expected to make a dominant contribution to elevated turbulent mixing and shear in west of Luzon Strait. We have strengthened the discussion on this in the revised text, see lines 246-275 and 422-431.

4. Discussion should be separated from Summary. The summary should summarize both results and a punch line of the discussion.
Responses:
We thank the reviewer for this good suggestion. We have separated the discussion from summary in the revised text.

Correction
We are very sorry for that there is a mistake in the magnitudes of dissipation spectra in Fig.2. We have corrected them in the revised text.

Our responses to Reviewer 1:

Note: The reviewer's original comments are in black, and our responses are in blue.

**General comments**

This is an interesting manuscript presenting a new data set observed in the South China Sea (SCS) by means of micro-structure shear (MSS) profilers. In four different geographical regions of the SCS, a total of 82 MSS profiles were obtained, covering the first 500m of the water column. Analyses mainly in terms of the dissipation rate and the diapycnal diffusivity are performed and discussed in comparison to the buoyancy and shear frequencies squared and the gradient Richardson number. Comparison of the results for the dissipation rate to the dissipation rate parameterizations by Gregg (1989) and to the parameterization by MacKinnon and Gregg (2005) is performed. In most situations, the latter model better represents the data. To explain this, internal wave spectra are derived from observations of 5 moorings in the SCS. Considerable deviations from the Garrett-Munk spectra, on which the Gregg (1989) model is based, explain the weakness of this model.

Having said this, the manuscript is generally publishable in Ocean Science. However, I have some concerns which need to be considered by the authors before acceptance can be recommended. Major revisions are required.

One concern is the lack of physical interpretation of the results. The authors should explain why certain areas show large or small shear and stratification, respectively. The role of high-amplitude internal waves entering from the Luzon Strait and their effect on mixing in the SCS needs to be discussed.

Responses:

Thank you for your good advice. There are a large amount of internal waves (tides) in the South China Sea, which has been reported in many literatures, such as Niwa and Hibiya (2004), Zhao et al. (2004), Klymak et al. (2006), Jan et al. (2007). Most of internal waves originate in Luzon Strait and propagate northwestwards through the deep water zone near Luzon Strait to the continental shelf (Alford et al. 2015). Mooring data (Lien et al. 2014) indicate that these internal waves would induce strong shear.

A comparison of the spatial distributions of turbulent mixing and internal waves indicated that the internal waves are expected to make a dominant contribution to elevate the turbulent mixing in west of the Luzon Strait. For reasons why strong shear and elevated dissipation occurred in west of the Luzon Strait, we have strengthened the discussion in the revised text, please see more detail in lines 246-275 and 422-431.

The study needs a better motivation. Which is the major knowledge gap to be filled? This should come out as a result from the introduction. In the moment it reads a bit like a report to present new data for the first time.

Responses:

We thank the reviewer for this good suggestion. There are two motivations for this study: firstly exploring the mixing
features and mixing regimes in different regions of the SCS and secondly assessing two parameterizations with the
microstructure data.

Many microstructure measurements have been conducted in the SCS. There is no doubt that these measurements have greatly aided our knowledge of turbulent mixing in the SCS. However, the microstructure measurements are localized and scattered with most of them focusing on the northern SCS. The mixing features and mixing regimes in different regions of
the SCS are still not fully understood. With the microstructure data in 2010, we present the spatial distribution of turbulent mixing in the upper ocean of the SCS and explore the mixing features and mixing regimes in different regions of the SCS in great detail. In the revised text, we strengthened the discussion on energy sources for the turbulent mixing (lines 246-275 and 422-431). Our observation indicated that strong turbulent mixing mainly occurred in west of the Luzon Strait where there are strong shear and weak stratification, and internal waves made a dominant contribution to the elevated turbulent mixing in
west of the Luzon Strait.

Another motivation for this paper is the assessment of two parameterizations (GH and MG models). Though many microstructure measurements have been conducted in the SCS, none of the two models has been assessed against the dissipation in the SCS. It remains unknown which parameterization can successfully reproduce the dissipation in the SCS and why. In manuscript, we assess the two parameterizations with the dissipation data of the SCS, which would provide
useful tools for ocean researchers. In fact, the microstructure measurements in the ocean are much fewer and more difficult than fine-structure measurements (i.e., CTD and ADCP measurements) in the ocean, especially in the deep sea. Thus to understand the spatial distribution and seasonal variation of the turbulent mixing in the ocean, researchers often turn to the parameterizations (Wu et al. 2011; Jing and Wu 2010). The assessment of parameterizations in the SCS would provide reference for researchers on the selection of parameterization to study the turbulent mixing in the SCS. The assessment of
parameterizations can also provide reference for modelers. Sea models have success in reproducing the water column structure in seasonally stratified shelf seas (Holt and Umlauf 2008; Simpson and Bowers 1981; Simpson and Hunter 1974). However, models need to calibrate a background mixing level for the correctly prediction (Rippeth 2005). The requirement of calibration reduces the success of models on shelf-wide scales. Before the water column structure in shelf seas can be modeled realistically, the distribution of mixing must be established and the major mixing processes identified and
parameterized. Parameterizations would provide a reference of the turbulence mixing for the modelers.

We have added the related content in the revised text as the reviewer suggested. Please see lines 52-74 in the revised text.

The shear estimated from 16m-bins is very coarsely resolved. Therefore, the gradient Richardson number calculated on that shear might be substantially underestimated. This needs to be discussed in more depth (not only in section 3.3).
Responses:

We understand the reviewer's concern. There is no doubt that the resolution of shear might affect the values of Richardson number. Unfortunately, we have only 16 m shear data, which prevents us from discussing the influence of shear resolution on the Richardson number. In our text, Richardson number was estimated following MacKinnon and Gregg (2005), see their Fig. 5. In the previous literatures, different shear resolutions were used to calculate the Richardson number, which range from 2 m to 16 m (MacKinnon and Gregg 2003; 2005; van der Lee and Umlauf 2011; Xie et al. 2013; Yang et al. 2014). High resolution of shear (2-4 m) was often used on the shelf area to catch the small scale internal waves. Low resolution of shear (8-16 m) was often used in deep water due to the large depth of the water column. Our observations mainly located in deep water, so the ADCP vertical resolution was set in 16 m to cover more depth. Although the resolution of 16 m might miss some overturning in our observation, it does not affect the comparison of Richardson number in different regions too much. We have strengthened the discussion on this in the revised text, see lines 177-183.

**Specific comments**

Line 24: wrong unit (should be m2/s)

Responses:

Thanks for reminding. We have corrected the mistake, see line 26 in the revised text.

Lines 37/38: "large numbers of … tides": better expression needed

Responses:

We have changed "large numbers of … tides" into "numerous … tides", see line 40 in the revised text.

Lines 53/54: not clear why these parameterisations are important for ocean models. Please explain, how those could be used, since I am not aware of an ocean model using these parameterisations. See also line 310, where something similar is postulated.

Responses:

We apologize for this confusion due to our inaccurate statement in the original text. Parameterization is important mainly due to its reference to modelers. Shelf sea models have success in reproducing the water column structure in seasonally stratified shelf seas (Holt and Umlauf 2008; Simpson and Bowers 1981; Simpson and Hunter 1974). However, models need to calibrate a background mixing level for the correctly prediction (Rippeth 2005). The requirement of calibration reduces the success of models on shelf-wide scales. Before the water column structure in shelf seas can be modeled realistically, the distribution of mixing must be established and the major mixing processes identified and parameterized. Parameterizations would provide a reference of the turbulence mixing for the modelers. Of course, we hope that these parameterizations can be applied to model in the near future.

Lines 54-56: here, a better motivation is needed.

Responses:

We have strengthened this in the revised text, see lines 52-74.

Line 58: what does "LT" stand for?

Responses:

We apologize for this confusing abbreviation. "LT" stands for "local time". We have changed "LT" into "local time" in the revised text, see lines 76-77.

Line 81: What is the detection limit for the TurboMAP profiler. You measure here very low dissipation rates of $10^{-10}$ W/kg. Are they still above the limit?

Responses:

The noise level of the TurboMAP profiler is $\varepsilon \sim 10^{-10}$ W/kg (Matsuno and Wolk 2005; Wolk et al. 2002). TurboMAP profiler resolves dissipation rates as low as $5 \times 10^{-10}$ W/kg. Lower values of dissipation rates can be inferred by comparing the measured spectra against the assumed universal form. We have added this in the revised text, see line 117.

Line 141: here the gradient Richardson number is defined. How is it calculated? Already here, and not as late as in section 3.3, you should discuss the consequences of a very low resolution shear estimate. Do also refer to the literature, how others cope with such low resolution of the shear when calculating Ri. I assume that at many other locations in your observations Ri<1/4 should occur (otherwise the disspation rate would be lower), but you do not resolved it.

Responses:

Richardson number was estimated following MacKinnon and Gregg (2005), see their Fig. 5. Firstly calculate 16-m shear and 2-m buoyancy frequency, then interpolate the 2-m buoyancy frequency to the 16-m shear grids, and then calculate the Richardson number with the shear and buoyancy frequency. In the previous literatures, different shear resolutions were used to in the calculation of the Richardson number, which range from 2 m to 16 m (MacKinnon and Gregg 2003; 2005; van der Lee and Umlauf 2011; Xie et al. 2013; Yang et al. 2014). High resolution of shear was often chosen (2-4 m) on the shelf area to resolve the small scale internal waves. Low resolution of shear was often chosen (8-16 m) in deep water due to the large depth of the water column. Our observations mainly located in deep water, so the ADCP vertical resolution was set in 16 m to cover more depth.

We agree with the reviewer that 16-m resolution might miss some overturning in our observation. Unfortunately, we have only 16-m shear data, which prevents us from discussing the influence of shear resolution on the Richardson number. In spite of this, it does not affect the comparison of Richardson number in different regions too much since the same shear resolution was used in the SCS. We have strengthened the discussion on this in the revised text, see lines 177-183.

Line 151: So, why is there strong shear and weak stratification in region 1 and vice versa in the other regions. Is it external and/or internal tides which are different across the SCS? Is it different wind regimes? In general, we need more physical oceanography here.

Responses:

We thank the reviewer for this good question. There are a large amount of internal waves (tides) in the South China Sea, which has been reported in many literatures, such as Niwa and Hibiya (2004), Zhao et al. (2004), Klymak et al. (2006), Jan et al. (2007). Most of internal waves originate in Luzon Strait and propagate northwestwards through the deep water zone near Luzon Strait to the continental shelf (Alford et al. 2015). Mooring data and microstructure measurements (Laurent 2008; Lien et al. 2014) indicate that internal waves would induce strong shear and produce elevated turbulence.

A comparison of the spatial distributions of turbulent mixing, winds, and internal waves indicated that elevated turbulent mixing in west of Luzon Strait (region 1) does not result from the effect of surface winds. The internal waves are expected to make a dominant contribution to elevated turbulent mixing and shear in west of the Luzon Strait. Unfortunately, we have only one profile of microstructure measurement and short time series (about one hour) of current velocity obtained by the ADCP for each station, thus it is impossible to separate the internal waves in various frequencies and explore their respective contributions to the dissipation. We have strengthened the discussion on this in the revised text, see lines 246-275.

Line 168: This overturn should have gone together with locally increased shear which is not resolved in the observations.

Responses:

We agree with the reviewer. Small overturning might be missed in our observation due to the low shear resolution.

Line 176. There is some confusion about the background value for eddy diffusivity in the ocean. In line 25, it is $10^{-5}$, in line 170, it is $5 \times 10^{-6}$, and here it is of the order of $10^{-6}$. These are considerably different values. Please clarify.

Responses:

We apologize for this confusion. Diapycnal diffusivity from turbulent mixing in the open ocean thermocline ranges from $5 \times 10^{-6}$ to $3 \times 10^{-5}$ $m^2/s$ (Gregg 1998; Polzin et al. 1995). We have clarified this in the revised text, see lines 26-27, 210, and 215-217.

Line 181: What is the physical meaning of depth and time averaged eddy diffusivity? Eddy diffusivity is a ratio (between flux and gradient), and the average of a ratio does not much sense to me. What is the additional information it gives in addition to the averaged dissipation rate (which makes sense)?

Responses:

In steady state, dissipation should equal the rate of transfer from the internal waves to turbulence mixing. Column integrated dissipation $\int_{-H}^{0} \rho \varepsilon dz$ (W/m$^2$) represents the rate of energy dissipation per square meter, which is often used to calculate the energy of internal waves losing to dissipation. Here, in order to compare the magnitude of dissipation and diffusivity in different stations and regions, we use the averaged dissipation rate, which can remove the influence of different thermocline depths on the comparison. Averaged eddy diffusivity and dissipation rate are also used to discuss the influence of wind and internal waves on the distribution of turbulence mixing. In addition, averaged eddy diffusivity is used to compare with the values in the open ocean.

Line 196: Does also the tidal phase at which the observations where taken matter? If not, why not? What about the wind forcing? Does it vary, does it matter? See Burchard & Rippeth (2009), where wind-induced shear across the thermocline matters. Burchard, H., and T.P. Rippeth, 2009: Generation of bulk shear spikes in shallow stratified tidal seas, J. Phys. Oceanogr., 39, 969-985.

Responses:

The barotropic tides extracted at location of $(18^o$N, $114^o$E) were used in the discussion without considering the tidal phase. The tidal phase does not affect our discussion too much because the bias in the arrival of spring-neap tides in different locations of the SCS is small. Fig A1 shows the time series of the barotropic tidal velocity predicted from TPXO 7.1 for three locations: $(21^o$N, $119^o$E), $(18^o$N, $114^o$E), and $(10^o$N, $114^o$E). One can see from Fig A1 that bias in the arrival of spring-neap tides between northern location $(21^o$N, $119^o$E) and southern location $(10^o$N, $114^o$E) was less than 3 hour. Related content has been added in the revised text, see lines 237-239.

We have added discussion on the effect of surface winds on turbulent mixing and shear in the revised text, see lines 246-265.

[Figure]

Fig. A1. Time series of the barotropic tidal velocity predicted from TPXO 7.1 for three locations: $(21^o$N, $119^o$E), $(18^o$N, $114^o$E), and $(10^o$N, $114^o$E).

Fig. 6b: wrong unit for eddy diffusivity. Equation after line 222: Something is wrong with the dimensions here. When f is 1/s, then $1.8\times10^{-6}$ should be $m^2/s^2$. Replace $1.8\times10^{-6}$ with a variable name and explain amount and unit in the text. Also, some of the brackets seem to denote an argument for a function $\cosh^{-1}$ and some denote a factor. Please clarify.

Responses:

Thanks for reminding. We have corrected the unit in Fig. 6. $\cosh^{-1}$ denotes inverse hyperbolic cosine function not a factor. The unit for $1.8\times10^{-6}$ is $m^2/s^2$. Actually a reference Coriolis frequency $f_0$ has been included in $1.8\times10^{-6}$. We have revised the manuscript according to the reviewers.

Line 222: Express cph also in Si units (1/s). Sometime cph and sometimes cpd is used, which I find confusing. The two parameterizations GH and MG should be explained for their physical reasoning. They are for different environments, deep ocean (GH) and shelf sea (MG), as I understand.

Responses:

We apologize for this confusion. 1 cph=$1.7\times10^{-3}$ s$^{-1}$ and 1 cpd=1 day$^{-1}$= $1.1574\times10^{-5}$ s$^{-1}$. To avoid confusion, we have changed the unit "cph" into "s$^{-1}$" and added "(1 cpd=1 day$^{-1}$)" in the revised text. Some of physical reasoning of the two parameterizations has been summarized in lines 358-367. For more information about the two parameterizations, one can refer to (MacKinnon and Gregg, 2003a). We are sorry for that we can't describe these two parameterizations better than them.

Lines 234/235: Are these data also for the thermocline region, or is it over the entire water column except for boundary layers?

Responses:

It is over the entire water column except for boundary layers.

Line 237: Here, a method for calculating Ri is explained. Is it different than before? Give this explanation at the first occurrence of Ri.

Responses:

The difference between the two is that 2-m buoyancy frequency was used at the first occurrence of Ri and 16-m buoyancy frequency was used at latter. We have clarified this in the revised text, see lines 178 and 309.

Fig. 7: Add locations (regions) to the plot.

Responses:

Thanks for reminding. Locations (regions) have been added to the plot.

Line 276: What is "fared"?

Responses:

"fared" means ''show''

Line 298: These different techniques and seasons should be discussed with respect to their effect on observed dissipation
rates.

Responses:

We thank the reviewer for this good suggestion. We have strengthened the discussion on this in the revised text, see lines 408-413.

Line 314: I had to look up the word eikonal. And it would be good, if the authors could briefly explain the eikonal model.

Responses:

Henyey et al. (1986) construct the analytic model by equating $\varepsilon$ to the net flux of energy passing out of the internal wave spectrum at large wave number, which they take as $2k_3^c$, corresponding to a 5-m vertical wavelength. Because the energy flows toward both lower and higher wave numbers,

$$\varepsilon_{HWF} = \frac{1-r}{1+r} \int_f^N \Phi_{flux}(2k_3^c, \omega)\, d\omega, \quad (1)$$

where $\Phi_{flux}(k_3, \omega)$ is the energy flux spectrum and $r$ is the ratio of the flux passing $2k_3^c$ toward lower wave numbers to the flux going toward higher wave numbers. The flux spectrum is formulated using ray-tracing equations to describe waves having high wave numbers propagating through a field in which most of the energy resides at low wave numbers. In keeping with a ray-tracing approach, $\Phi_{flux}(k_3, \omega)$ can be expressed as the product of the GM model energy spectrum $\Phi_E(k_3, \omega)$ and the rate at which the vertical wave number changes as the packet passes through the spectrum,

$$\Phi_{flux}(k_3, \omega) = \Phi_E(k_3, \omega) \left| \frac{dk_3(\omega)}{dt} \right| \text{ and} \quad (2)$$

$$\Phi_E(k_3, \omega) = \frac{2b^3 \beta_* N N_0 E_{GM}}{\pi (\beta_* + \beta)^2} \frac{f}{\omega(\omega^2 - f^2)^{1/2}}, \quad (3)$$

where $\beta_* = j_* \pi N / N_0$ and $\beta = bk_3$. Because the flux is evaluated at a wave number in the roll-off region of the energy spectrum,

$$\Phi_E(k_3, \omega) = \frac{k_3}{k_3^c} \Phi_E(k_3^c, \omega) \quad k_3 > k_3^c, \quad (4)$$

where $k_3^c = (3 Ri_c j_* b E_{GM})^{-1}$, For large wave numbers, the ray-tracing equations give

$$\left| \frac{dk_3}{dt} \right| = |\mathbf{S} \cdot \mathbf{k}|, \quad (5)$$

where $\mathbf{S}$ is the shear vector. If $\mathbf{S}$ and $\mathbf{k}$ are uncorrelated,

$$\left| \frac{dk_3}{dt} \right| = Nk_h \left[ \frac{1}{2} \langle Ri^{-1} \rangle \right]^{1/2}. \quad (6)$$

where $k_h = k_3 [(\omega^2 - f^2)/(N^2 - \omega^2)]^{1/2}$. In evaluating $\langle Ri^{-1} \rangle$, they take account of the additional shear contributed by wave numbers between $k_3^c$ and $2k_3^c$,

$$\langle Ri^{-1} \rangle = Ri_c^{-1} [1 + \ln(k_3/k_3^c)]. \quad (7)$$

Using (2)-(7) in (1) leaves the integral

$$\int_f^N \omega^{-1} (N^2 - \omega^2)^{-1/2}\, d\omega = N^{-1} \cosh^{-1}(N/f)$$

Following Munk (1981) $Ri_c^{-1}$=0.5, $k_3/k_3^c = 2$, and r =0.4,

$$\varepsilon_{HWF} = 0.33 f^{-1} [4\pi^{-1} j_* b E_{GM} f]^2 \cosh^{-1}(N/f).$$

Line 327: typo "flied".

Responses:

Thanks for reminding. We have corrected the typo "flied".

Line 330: Add "tidal" in front of "frequencies".

Responses:

Thanks for reminding. We have added the missing word "tidal".

Line 332: What are the $D_3$, $D_4$ and $D_5$ frequencies?

Responses:

$D_3$, $D_4$ and $D_5$ are the higher tidal harmonic frequencies, i.e., $D_3=D_1+D_2$, $D_4=D_2+D_2$, and $D_5=D_2+D_3$, where $D_1$ and $D_2$ represent the diurnal and semidiurnal tidal frequencies, respectively. These higher tidal harmonic frequencies mainly result from nonlinear interaction between internal waves [van Haren, 2002; van Haren, 2003; Xie, 2010]. We have cited the work of van Haren (2003), van Haren et al. (2002), and Xie et al. (2010) in the revised text, see lines 377-379.

Line 335/336: Sentence is a repetition of what has been written further up.

Responses:

Thanks for reminding. We have deleted repetition "The GH model is typically evaluated for the wave field with the GM

spectral shape".

Line 340/341: How can a model for bulk averages be used for constructing profiles (such as in fig.5)?

Responses:

With the calculated shear $S(z)$ and buoyancy frequency $N(z)$, profiles can be constructed from the equations of $\varepsilon_{MG}(z) =$

$\varepsilon_0 \frac{N(z)}{N_0} \frac{S(z)}{S_0}$ and $\varepsilon_{GH} = 1.8 \times 10^{-6} \left[ f \cosh^{-1}\left(\frac{N_0}{f}\right) \right] \left[ \frac{S(z)^4}{S_{GM}^4} \right] \left[ \frac{N(z)^2}{N_0^2} \right]$.

Line 345: word missing after "observed".

Responses:

Thanks for reminding. We have added the missing word "dissipation".

Our responses to Reviewer 2:

Note: The reviewer's original comments are in black, and our responses are in blue.

The authors conducted an extensive field campaign to survey turbulence in the South China Sea and reported a spatial pattern of turbulent intensity from the observed data. They also compared the observed data against two theoretical models. The results are well known. I found no new information. I appreciate that the amount of work involved in the data collection, but as far as the science concerns the present manuscript reads like a data report. I found no new scientific finding and no new facts other than the survey was conducted in the South China Sea. Unless Ocean Science accept a manuscript aimed at a data report, I would not recommend this manuscript for an official scientific paper. For their revision purpose I will comment on this manuscript as followed:

Responses:

We thank the reviewer for the comment. In this paper, the turbulent mixing in the South China Sea (SCS) is analysed from two aspects, not just a simple data report. Firstly we explore the mixing features and mixing regimes in the SCS in great detail. Secondly we assess two parameterizations with the microstructure data.

Many microstructure measurements have been conducted in the SCS. There is no doubt that these measurements have greatly aided our knowledge of turbulent mixing in the SCS. However, the microstructure measurements are localized and scattered with most of them focusing on the northern SCS. The mixing features and mixing regimes in different regions of the SCS are still not fully understood. With the microstructure data in 2010, we present the spatial distribution of turbulent mixing in the upper ocean of the SCS and explore the mixing features and mixing regimes in different regions of the SCS. In the revised text, we strengthened the discussion on energy sources for the turbulent mixing (lines 246-275 and 422-431). Our observation indicated that strong turbulent mixing mainly occurred in west of the Luzon Strait where there are strong shear and weak stratification, and internal waves made a dominant contribution to the elevated turbulent mixing in west of the Luzon Strait.

Another work in this paper is the assessment of two parameterizations (GH and MG models). Though many microstructure measurements have been conducted in the SCS, none of the two models has been assessed against the dissipation in the SCS. It remains unknown which parameterization can successfully reproduce the dissipation in the SCS and why. In manuscript, we assess the two parameterizations with the dissipation data of the SCS, which would provide useful tools for ocean researchers. In fact, the microstructure measurements in the ocean are much fewer and more difficult than fine-structure measurements (i.e., CTD and ADCP measurements) in the ocean, especially in the deep sea. Thus to understand the spatial distribution and seasonal variation of the turbulent mixing in the ocean, researchers often turn to the parameterizations (Wu et al. 2011; Jing and Wu 2010). The assessment of parameterizations can also provide reference for modelers. Models often need to calibrate a background mixing level to correctly predict the physical phenomenon (Rippeth 2005). The requirement of calibration reduces the success of models on large scales since differing forcing mechanisms and mixing processes require specific methods and levels of tuning. Before the physical phenomenon can be modeled realistically, the distribution of mixing must be established and the major mixing processes parameterized. Parameterizations can provide a reference of the background mixing for the modelers.

According to reviewers' comments, we have strengthened the content of motivation (lines 52-74) and added discussion on the effect of surface winds and internal waves on turbulent mixing and shear (lines 246-275) in the revised text.

1. Material and methods should be explained in more detail. Also those mooring data outside the observation period should be removed from the text. The Luzon Strait is not well known to the audience. Indicate where LS is.

Responses:

We thank the reviewer for this suggestion. Mooring data were used to show the feature of wave field in the SCS. We think
that it should be kept in the text. We have strengthened the description of the material and methods in the revised text (lines 76-123). The location of Luzon Strait has been indicated in Fig .1.

2. P.4 line83: "caused by instrument vibrations" If so, you can verify the vibration with accelerometer data. Those are mostly electronic noise.
Responses:

We apologize for this confusion due to our inaccurate statement in the original text. They are vibration noise caused by the strumming of the suspension wires in the flow (Wolk et al. 2002). We have clarified in the revised text, see lines 108-109.

3. You have to focus the science. What is deriving high turbulence in Region 1. Most likely internal tides are playing a major
role in generation of turbulence.

Responses:

We thank the reviewer for this good question. There are a large amount of internal waves (tides) in the SCS, which has been reported in many literatures, such as Niwa and Hibiya (2004), Zhao et al. (2004), Klymak et al. (2006), Jan et al. (2007). Most of internal waves originate in the Luzon Strait and propagate northwestwards through the deep water zone near Luzon
Strait to the continental shelf (Alford et al. 2015). Mooring data (Lien et al. 2014) indicate that internal waves would induce strong shear. A comparison of the spatial distributions of turbulent mixing, winds, and internal waves suggests that the elevated turbulent mixing and shear in west of the Luzon Strait (region 1) does not result from the effect of surface winds. The internal waves are expected to make a dominant contribution to elevated turbulent mixing and shear in west of Luzon Strait. We have strengthened the discussion on this in the revised text, see lines 246-275 and 422-431.

4. Discussion should be separated from Summary. The summary should summarize both results and a punch line of the discussion.

Responses:

We thank the reviewer for this good suggestion. We have separated the discussion from summary in the revised text.

Correction

We are very sorry for that there is a mistake in the magnitudes of dissipation spectra in Fig.2. We have corrected them in the revised text.

Note that the modifications in marked-up manuscript version are marked in red.

---

## Referee Report (RR1)

2nd Review on "Spatial distribution of turbulent mixing in the upper ocean of the South China Sea" (os-2016-80) by X.D. Shang, C.R. Liang and G.Y Chen.

**General**

The authors have improved the manuscript based on the comments made by two reviewers. However, I still hold the same opinion that I found no new scientific finding and no new facts other than the survey was conducted in the South China Sea.

In fact, the authors mention that the objectives of this study are the following two points:

1) We explore the mixing features and mixing regimes in the SCS.

2) None of two models (GH and MG) has been assessed against the dissipation in the SCS.

Cleary this work is focusing on a local area, not a general scientific question. Even the work is a local study, at least a scientific question should be identified. The important features the authors found were:

1) Mixing is strong in the northern part of the SCS. The mixing is induced by internal tides.

2) MG fits the observed mixing intensity well.

As they state in Summary, there are numerous previous works regarding mixing in the SCS. This is also mentioned in Introduction. Do not repeat the same context. This part should be move to Introduction so that what have been identified and what has not been studied in the SCS. Then states, the motivation of this study focusing on the following two points:

1) Where is the strong mixing taking place in the SCS? And what derives the mixing?

2) In order to estimate the mixing without microstructure measurements, we explore the two models. And investigate which one works better and why it works better.

At a moment, Introduction ends with sort of Summary. Do not mention the conclusion in Introduction. Once the authors resort the context and slim down the whole paper without stating the same context, the manuscript should become acceptable. Next I will state specific comments.

**Specific**

p.14,L43-44: "almost two orders" No, at most one order!

p.15, L73: "the improved ocean model" You are not improving the model. Do not state your wish list.

p.20, L167, 171: $S^2$ and $N^2$ should be in italic. Elsewhere as well.

p.21, Fig.5 caption: "buoyancy frequency" -> "buoyancy frequency squared"

p.22,L222: $z_b$ and $z_t$ should be defined where they are first appeared (L159 and 160).

p.22,L224: Before Fig.6 appears, show the probability density function (pdf) of epsilon for each region estimated from non-parametric pdf estimator, such as histogram. The pdf should suggest you whether the source of mixing is a single forcing (single mode) or multiple forcing (multiple modes).

It is important to investigate the pdf when you discuss statistics, even the mean. If you look at the correlation between tidal elevation and mean dissipation rate, you might find some positive correlation. Later part, you pointed out that the source of mixing is due to internal waves, most likely internal tides. Internal tides have to generated from some forcing, usually barotropic tides. Both K1 and M2 components can propagate at this latitude, so may not be easy to identify which one is dominating the most. But maybe previous studies, such as Zhao et al. (2004) mentioned the properties of the internal tides. Also some numerical works have been done in this area, that should be useful information to state which one is dominating.

p.24,L266: "Internal wave" should be "Internal tide".

p.24, L274: "These observations indicate" No, you are not demonstrating if it is due to internal wave (tide). Say "suggested" instead of "indicate".

p.25,L288-289: "does not result from the effect of spring-neap tides" Are you sure they are uncorrelated? If so, what generates Internal waves (tides). Usually internal tide (baroclinic mode) is generated from surface tide (barotropic mode).

p.26, Fig.7: In addition to Fig.7, I would like to see a direct comparison of observed epsilon vs model epsilon and show $R^2$ (not correlation coefficient). And test the linear regression equation. You might claim that Fig 5 does it.

But this not sufficient. You have to test it. You must also discuss the error band based on the model. You promised to deliver a useful tool to modelers that can assess the mixing intensity without microstructure data. So you need to state the accuracy of the model.

p.29,Fig. 9: These diagrams are all based on averaged velocity between 60 and 270 m. Are you looking at only barotropic components? How could you be make sure you can identify both barotropic and baroclinic component separately? Since mixing (turbulence) is most likely generated from internal tides (K1 and M2), it is important to show two components separately.

p.29-30: The first paragraph should be in Discussion. This is not a summary of your work. You must clarify what have been reported from the previous studies and show what this study identified. A part of this story may be in Introduction as a motivation of this study.

---

## Referee Report (RR2)

Third review on "Spatial distribution of turbulent mixing in the upper ocean of the South China Sea" (os-2016-80) by X.D. Shang, C.R. Liang and G.Y Chen.

The authors have improved the manuscript based on the previous comments. I can recommend publication after the following points are clarified:

1) Line 286-294: This should be removed since the same context is repeated in Summary.

2) Line 223: PDF estimator -> PDF estimator (histogram)

3) Fig.10: Theta is not a test statistic. The linear regression should be tested with t-test. Also $R^2$ needed to be mentioned. $R^2$ (coefficient of determination) is a standard variable defined in the most statistical book. See the following site as an example:

http://www.r-tutor.com/elementary-statistics/simple-linear-regression/coefficient-det ermination

Also this regression equation should be used to predict accurate dissipation rate from the model.

4) Line 385: What parameterization?

---

## Author Response (AR2)

**Our responses to Reviewers:**

**Note: The reviewer's original comments are in black, and our responses are in blue.**

**Reviewer comments (A) with <>**

5 **General**

The authors have improved the manuscript based on the comments made by two reviewers. However, I still hold the same opinion that I found no new scientific finding and no new facts other than the survey was conducted in the South China Sea. <>

In fact, the authors mention that the objectives of this study are the following two points:

10 1) We explore the mixing features and mixing regimes in the SCS.

2) None of two models (GH and MG) has been assessed against the dissipation in the SCS.

Clearly this work is focusing on a local area, not a general scientific question. Even though the work is a local study, at least a scientific question should be identified. <> The important features the authors found were:

1) Mixing is strong in the northern part of the SCS. The mixing is induced by internal tides.

15 2) MG fits the observed mixing intensity well.

As they state in Summary, there are numerous previous works regarding mixing in the SCS. This is also mentioned in Introduction. Do not repeat the same context. This part of the Summary should be move to Introduction to clarify what has been identified and what has not been studied in the SCS. Then state the motivation of this study focusing on the following two points:

20 1) Where is the strong mixing taking place in the SCS? And what drives the mixing?

2) In order to estimate the mixing without microstructure measurements, we explore the two models. And investigate which one works better and why it works better.

At the moment, Introduction ends with a sort of Summary. Do not mention the conclusion in the Introduction. Once the authors resort the context and slim down the whole paper without stating the same context, the manuscript should become

25 acceptable. <>

Responses:

We are very sorry for the bad structure of our manuscript and thank you for your good advice. We have re-sorted the manuscript as the reviewer suggest. We removed the conclusion in the Introduction and stated the motivation of our study in the revised text (lines 54-59). A sort of Summary was also added in the end of the Introduction (lines 72-75).

**Next I state specific comments.**

**Specific**

p.14,L43-44: "almost two orders" No, at most one order!

Responses:

35    Thanks for reminding. We have corrected the mistake, see line 43 in the revised text.

p.15, L73: "the improved ocean model" You are not improving the model. Do not state your wish list.

Responses:

Thank you for your advice. We have removed "improved" in the revised text, see line 71.

p.20, L167, 171: $S^2$ and $N^2$ should be in italic. Elsewhere as well.

40    Responses:

We have revised the manuscript as the reviewer suggested.

p.21, Fig.5 caption: "buoyancy frequency" -> "buoyancy frequency squared"

Responses:

Thanks for reminding. We have corrected the mistakes throughout the manuscript.

45    p.22,L222: $z_b$ and $z_t$ should be defined where they first appear (L159 and 160).

Responses:

We have defined $z_b$ and $z_t$ where they first appear in the revised text (lines 157-160).

p.22,L224: Before Fig.6 appears, show the probability density function (pdf) of epsilon for each region estimated from non-parametric pdf estimator, such as histogram. The pdf should suggest whether the source of mixing is a single forcing (single

50    mode) or multiple forcing (multiple modes).

It is important to investigate the pdf when you discuss statistics, even the mean. If you look at the correlation between tidal elevation and mean dissipation rate, you might find some positive correlation. Later, you pointed out that the source of mixing is due to internal waves, most likely internal tides. Internal tides have to generated from some forcing, usually barotropic tides. Both K1 and M2 components can propagate at this latitude, so may not be easy to identify which one is

55    dominating. But maybe previous studies, such as Zhao et al. (2004) mentioned the properties of the internal tides. Also some numerical works have been done in this area, that should be useful information to state which one is dominating.

Responses:

Thank you for your good advice. We have added the probability density function (pdf) of epsilon for each region and strengthened the discussion on this part in the revised text (lines 217-227). Both the diurnal and semidiurnal internal tides

60    generated in the Luzon Strait can propagate westward into the South China Sea and the diurnal internal tide is the dominating one [Zhao, 2014]. In addition to the diurnal and semidiurnal internal tides, other internal waves with higher frequency are also generated near the Luzon Strait[Alford et al., 2015; Lien et al., 2014; Xie et al., 2011; Zhao et al., 2004]. They can also contribute to the turbulent mixing. Due to our limited data, we cannot determine which frequency internal wave makes the dominant contribution to the turbulent mixing. We think that further observations with frequent

65    microstructure measurements and long time-series of current velocity measurements are needed to identify the dominant mixing mechanism in the northern SCS.

p.24, L266: "Internal wave" should be "Internal tide". <>

Responses:

We understand the reviewer's concern. "Internal wave" in the manuscript is a general term, which includes the internal tides.

Actually internal tides are internal waves at a tidal frequency. In the northern South China Sea, not only internal waves with tidal frequency but also other internal waves with high frequency are generated [Alford et al., 2015; Lien et al., 2014; Xie et al., 2011]. In order to avoid confusion, we have changed the "internal wave" into "internal waves and internal tides" in the revised text, see lines 17, 275, 293, and 443.

p.24, L274: "These observations indicate" No, you are not demonstrating if it is due to internal wave (tide). Say "suggested" instead of "indicate".

Responses:

Thanks for reminding. We have changed "indicate" into "suggested ", see line 284 in the revised text.

p.25, L288-289: "does not result from the effect of spring-neap tides" <> Are you sure they are uncorrelated? If so, what generates Internal waves (tides). Usually internal tide (baroclinic mode) is generated from surface tide (barotropic mode).

Responses:

We apologize for this confusion due to our inaccurate statement in the original text. What we want to express is that the mixing pattern in the SCS with elevated turbulent mixing concentrated in region 1 does not result from the different measure time in different stations (spring-neap tides). Spring-neap tides do affect the turbulent mixing. Strong turbulent mixing generally occurs during spring tides [Peters and Bokhorst, 2000]. Thus it is possible that the microstructure measurements in region 1 were taken during spring tides and microstructure measurements in regions 2-4 were taken during neap tides, and the elevated turbulent mixing in region 1 may result from different measure time. To rule out this possibility, we compare the spring-neap tides with the dissipation. We find that not all of microstructure measurements in region 1 are taken during spring tides, and even at stations in region 1 conducted during neap tides the turbulent mixing are still stronger than that of the stations in regions 2-4 conducted during spring tides. Thus the elevated turbulent mixing in region 1 does not result from the measure time. In order to avoid this confusion, we have reorganized in this part in the revised text, see lines 239-250.

p.26, Fig.7: In addition to Fig.7, I would like to see a direct comparison of observed epsilon vs. model epsilon and show $R^2$ (not correlation coefficient). And test the linear regression equation. You might claim that Fig 5 does it. But this not sufficient. You have to test it. You must also discuss the error band based on the model. You promised to deliver a useful tool to modelers that can assess the mixing intensity without microstructure data. So you need to state the accuracy of the model.

Responses:

We have shown a direct comparison of observed epsilon vs. model epsilon in the revised text. Error band is also discussed (lines 359-378). We are sorry for that we have some difficulty to understand what $R^2$ is.

p.29,Fig. 9: These diagrams are all based on averaged velocity between 60 and 270 m. <> Are you looking at only barotropic components? How could you be make sure you can identify both barotropic and baroclinic component separately? Since mixing (turbulence) is most likely generated from internal tides (K1 and M2), it is important to show two components separately.

Responses:

We are sorry for that we cannot separate the barotropic component from the baroclinic component since we have no full-depth velocity. Both the diurnal and semidiurnal internal tides may contribute to the turbulent mixing. However, we cannot explore their respective contributions to the dissipation with our limited data.

p.29-30: The first paragraph should be in Discussion. This is not a summary of your work. You must clarify what have been reported from the previous studies and show what this study identified. A part of this story may be in Introduction as a motivation of this study.

Responses:

Thank you for your good advice. We have restructured our manuscript and moved the first paragraph in Summary to the Discussion (lines 380-389).

Responses:

Thanks for reminding. We have corrected the mistakes throughout the revised text.

Line 39. Omit "numerous"?

Responses:

Thanks for reminding. We have omitted "numerous" in the revised text (line 39).

Line 48. "over the shelf break"

Responses:

Thanks for reminding. We have changed "in the shelf break" into "over the shelf break ", see line 47 in the revised text.

Line 68. Simpson and Hunter 1974 is about stratification but they did not have a high-resolution model; if you wish to cite their paper you need to relate it only to the factors giving summer stratification.

Responses:

Thanks for reminding. We have removed the reference of Simpson and Hunter 1974, see lines 65-66 in the revised text.

Lines 76-77. Better ". . performed from 26 April . . (local time) prior to the South China Sea summer monsoon . . . onset. A total . ."

Responses:

Thank you for your advice. We have changed the structure of the sentence as the reviewer suggested; see lines 77-78 in the revised text.

Line 117. "and the length of the profiler itself". I think you might mean "and the tendency for the profiler to follow the larger-scale flow".

Responses:

Yes. The larger-scale flow also influences the profiler.

Line 225. "displayed a decreasing trend" -> "decreased"?

Responses:

Thanks for reminding. We have changed "decreasing" into "decreased", see line 231 in the revised text.

Lines 259-261. These sentences repeat earlier text.

Responses:

Thanks for reminding. We have refined these sentences; see lines 263-265 in the revised text.

Equations before line 295. $\varepsilon$ (dissipation) has different dimensions from f. Therefore $1.8\times10^{-6}$ has dimensions and the value depends on the units used. You need to state these (as referee 1 asked). Also shear has dimensions so $1.66 \times10^{-10}$ has dimensions and the value depends on the units used.

Responses:

We have revised the text as the reviewer suggested; see lines 299-303 in the revised text.

Line 322. Better ". . display a pattern qualitatively consistent with . .".

Responses:

Thanks for reminding. We have changed the structure of the sentence as the reviewer suggested; see lines 326 in the revised text.

Line 373. "flied" -> "field".

Responses:

Thanks for reminding. We have corrected the mistake, see line 410 in the revised text.

Line 376. Better "inertial (f) and tidal frequencies (diurnal O1 and K1; semidiurnal M2); the peaks imply that energy . ."

175 Responses:

Thanks for reminding. We have changed the structure of the sentence as the reviewer suggested; see line 413 in the revised text.

Line 378. I think you should add "(respectively about 3, 4, 5 cycles per day)" after "D5"; a referee asked for explanation.

Responses:

180 Thanks for reminding. We have added "(respectively about 3, 4, 5 cycles per day)" into the revised text; see line 413 in the revised text.

Line 413. ". . diffusivity to vary." ?

Responses:

We have removed this sentence, which does not influence our discussion, see line 389 in the revised text.

185 Line 416. ". . in these conditions and . ."

Responses:

Thanks for reminding. We have corrected the mistake, see line 435 in the revised text.

Line 441. ". . observed dissipation would change . ."?

Responses:

190 Yes. We have added "dissipation" into the revised text; see line 459 in the revised text.

Note that the modifications in marked-up manuscript version are marked in red

[revised manuscript text omitted]

---

## Author Response (AR3)

**Our responses to Reviewers:**

**Note: The reviewer's original comments are in black, and our responses are in blue.**

**Reviewer comments**

The authors have improved the manuscript based on the previous comments. I can recommend publication after the following points are clarified:

1) Line 286-294: This should be removed since the same context is repeated in Summary.

Responses:

Thank you for your advice. We have removed the repeated context in the revised text, see line 286.

2) Line 223: PDF estimator -> PDF estimator (histogram)

Responses:

Thanks for reminding. We have added "(histogram)" in the revised text (line 224).

3) Fig.10: Theta is not a test statistic. The linear regression should be tested with t-test. Also $R^2$ needed to be mentioned. $R^2$ (coefficient of determination) is a standard variable defined in the most statistical book. See the following site as an example: http://www.r-tutor.com/elementary-statistics/simple-linear-regression/coefficient-determination. Also this regression equation should be used to predict accurate dissipation rate from the model.

Responses:

Thank you for your advice. We have strengthened the discussion on this part in the revised text, see lines 355-371. In Fig.10, we have changed the x-axis into modeled dissipation and the y-axis into observed dissipation.

4) Line 385: What parameterization?"

Responses:

The parameterization they used was Gregg–Henyey–Polzin parameterization, similar to the GH model. We have added "Gregg–Henyey–Polzin" in the revised text (lines 378).

**Editor's comments**

Firstly, regarding the referee comments

1) I agree. But in the Summary, about line 437, you also need to mention region 2.

Responses: We have removed the repeated context and mentioned region 2 in the Summary in the revised text, see line 286 and 430.

3) I agree. The point is that the variance of the data from the line fitted by regression, i.e. "residual variance", should be significantly less than the original data variance. $R^2$ is the proportion of variance "explained" by the regression. $1-R^2$ is the reduction factor {(residual variance)/(original data variance)}. This is similar to but not the same as your θ; in both cases, the smaller the better. Statistical tests concern whether the residual variance is small enough that the coefficient of regression is significantly different from zero. An important factor in significance is how many independent data points there are in the regression. I am not an expert in all this and suggest that you look at the Web site suggestion by the referee.

Responses: We have strengthened the discussion on this part in the revised text as the reviewer suggested, see lines 355-371. In Fig.10, we have changed the x-axis into modeled dissipation and the y-axis into observed dissipation.

Please also address referee comments 2) and 4). Regarding comment 4) you may wish to label the equations after lines 298 and 301 if you want to specify "parameterizations" later.

Responses:

We have revised the text as the reviewer suggested; see lines 286 and 378 in the revised text.

Line 15. Better ". . , induced by shear instability, . ." (add "," twice)

Responses:

We have revised the text as the editor suggested; see line 15 in the revised text.

Line 28. Better ". . concentrated over rough topography [ . ."

Responses:

Thank you for your advice. We have revised the text as the editor suggested; see line 28 in the revised text.

Line 42. Better ". . the Pacific."

Responses:

Thank you for your advice. We have revised the text as the editor suggested; see lines 42-43 in the revised text.

Lines 56. Better ". . we present direct microstructure measurements that cover . ."

Responses:

Thank you for your advice. We have revised the text as the editor suggested; see lines 55-56 in the revised text.

Line 57 "In addition . ." to line 72 ". . mixing level." I think this is mostly motivation for parameterization and would be better as a separate paragraph. Also the second sentence "In order to estimate . . better." might best be at the end of this separate paragraph.

Responses:

Thank you for your advice. We have revised the text as the editor suggested; see lines 71-73 in the revised text.

Line 71. ". . dependent on an ocean turbulence model . ."

Responses:

Thank you for your advice. We have revised the text as the editor suggested; see line 70 in the revised text.

Line 117. ". . and the length of the profiler itself." After "itself" insert "as the profiler tends to follow the larger-scale flow"?

Responses:

Thank you for your advice. We have revised the text as the editor suggested; see line 117 in the revised text.

Line 121. Best to define the buoyancy frequency N here by $N^2 = - g/\rho \ \partial\rho/\partial z$.

Responses:

Thank you for your advice. We have revised the text as the editor suggested; see line 119 in the revised text.

Lines 122-123. Better ". . are the respective zonal and meridional . . ADCP. The mean . ."

Responses:

Thank you for your advice. We have revised the text as the editor suggested; see lines 122-123 in the revised text.

Figure 4. The grey curves are rather faint and thin; they do not show well on a printed version.

Responses:

Thank you for your reminding. We have changed the thin curves into bold curves in Figure 4.

Lines 183-185. The sentence "Another . . in the water column." does not seem to add to what was said in lines 173-175. Or do you need to replace "in the water column" (which is anywhere) with "below the surface mixed layer", "in the thermocline" or "below the thermocline" (for example).

Responses:

Thank you for your advice. We have the sentence "Another . . in the water column." into "One prominent feature . . below the surface mixed layer."; see lines 184-185 in the revised text.

Line 231. ". . $<\varepsilon>_T$ displayed a decreased trend towards . ." This means that the trend changed. Do you mean that or just that

$<\varepsilon>_T$ decreased, i.e. ". . $<\varepsilon>_T$ decreased towards . ."

Responses:

We mean ". . $<\varepsilon>_T$ decreased towards . ." . We have revised the text; see line 232 in the revised text.

Line 239. Better ". . different times and the measurement time . ."

Responses:

Thank you for your advice. We have revised the text as the editor suggested; see line 240 in the revised text.

Lines 242, 248, 443. "measure" -> "measurement"

Responses:

We have revised the text as the editor suggested; see lines 244, 249, and 435 in the revised text.

Line 394. Not "predication". Maybe "prediction"? Or ". . It is typically evaluated . ."

Responses:

Thank you for your advice. We have revised the text as the editor suggested; see line 387 in the revised text.

Note that the modifications in marked-up manuscript version are marked in red.

[revised manuscript text omitted]

---

## Author Response (AR4)

**Our responses to Editor**

**Note: The Editor's original comments are in black, and our responses are in blue.**

Lines 44-45 "the fine-scale parameterization". If this was the same parameterization as in Tian et al. (2009), then this is OK. Otherwise, "a" not "the".

5 Responses:

Thanks for reminding. We have changed "the" into "a" in the revised text (lines 44-45).

Line 62. Omit "the".

Responses:

10 Thanks for reminding. We have omitted "the" in the revised text (lines 62).

Line 107. "Figs. 2b-2g"

Responses:

Thanks for reminding. We have corrected the mistake in the revised text (lines 107).

Line 119. "-g/ρ ∂ρ/∂z" Either brackets (. .) around g/ρ or no spaces in g/ρ and good space before ∂ρ/∂z as written here.

Responses:

Thanks for reminding. We have added brackets (. .) around g/ρ in the revised text (lines 119).

20 Line 183 "underestimated". At present this sentence means "Richardson number . . might be underestimated". However, I suspect you want to say shear might be underestimated, hence Richardson number overestimated and possibility of instability underestimated.

Responses:

Thanks for reminding. What we want to express is what you say. We have changed "underestimated" into "overestimated" in

25 the revised text (lines 183).

Line 253. "and" not "which" (which refers to the item immediately before; here you want to refer to the wind).

Responses:

Thanks for advice. We have changed "which" into "and" in the revised text (lines 253).

30

Line 276. ". . internal tides are candidates .." (omit "the").

Responses:

Thanks for reminding. We have omitted "the" in the revised text (lines 276).

35   Line 366. "larger" not "smaller" (larger is better!)

Responses:

Thanks for reminding. We have corrected the mistake in the revised text (lines 366).

Lines 385-386. I think ". . stationary; the energy . ." is better. Otherwise, "maintain" -> "maintaining" on line 386.

40   Responses:

Thanks for advice. We have changed "maintain" into "maintaining" in the revised text (lines 386).

45

50

55

60

Note that the modifications in marked-up manuscript version are marked in red.

[revised manuscript text omitted]